# SorCS1-mediated sorting in dendrites maintains neurexin axonal surface polarization required for synaptic function

**Luís F. Ribeiro**[1,2], **Ben Verpoort**[1,2], **Julie Nys**[1,2], **Kristel M. Vennekens**[1,2], **Keimpe D. Wierda**[1,2], **Joris de Wit**[1,2]*

**1** VIB Center for Brain & Disease Research, Herestraat, Leuven, Belgium, **2** KU Leuven, Department of Neurosciences, Leuven Brain Institute, Herestraat, Leuven, Belgium

* joris.dewit@kuleuven.vib.be

**Data Availability Statement:** All relevant data are within the paper and its Supporting Information files.

## Abstract

The pre- and postsynaptic membranes comprising the synaptic junction differ in protein composition. The membrane trafficking mechanisms by which neurons control surface polarization of synaptic receptors remain poorly understood. The sorting receptor Sortilin-related CNS expressed 1 (SorCS1) is a critical regulator of trafficking of neuronal receptors, including the presynaptic adhesion molecule neurexin (Nrxn), an essential synaptic organizer. Here, we show that SorCS1 maintains a balance between axonal and dendritic Nrxn surface levels in the same neuron. Newly synthesized Nrxn1α traffics to the dendritic surface, where it is endocytosed. Endosomal SorCS1 interacts with the Rab11 GTPase effector Rab11 family-interacting protein 5 (Rab11FIP5)/Rab11 interacting protein (Rip11) to facilitate the transition of internalized Nrxn1α from early to recycling endosomes and bias Nrxn1α surface polarization towards the axon. In the absence of SorCS1, Nrxn1α accumulates in early endosomes and mispolarizes to the dendritic surface, impairing presynaptic differentiation and function. Thus, SorCS1-mediated sorting in dendritic endosomes controls Nrxn axonal surface polarization required for proper synapse development and function.

## Introduction

Neurons are highly compartmentalized cells that need to maintain distinct membrane identities. This is especially clear at the synapse, where pre- and postsynaptic membranes differ dramatically in their protein composition [1]. Membrane trafficking mechanisms that organize and maintain the polarized distribution of receptor proteins during the formation, maturation, and plasticity of synapses are of key importance for the proper function of neural circuits [2,3].

Neurexins (Nrxns), expressed from 3 genes as α-, β-, and γ-Nrxns, are essential presynaptic adhesion molecules that engage in a network of interactions with multiple pre- and postsynaptic extracellular ligands [4] and act in a cell-type–specific manner to regulate synapse number, function, and plasticity [5–9]. Mutations in *NRXNs* have been associated with multiple

**Funding:** L.F.R. is supported by Marie Sklodowska-Curie postdoctoral fellowship H2020-MSCA-IF-2014 (https://ec.europa.eu/research/mariecurieactions/actions/individual-fellowships_en) and Flanders Research Organization (FWO) Postdoctoral fellowship 12N0316N/12N0319N (https://www.fwo.be/en/fellowships-funding/postdoctoral-fellowships/). B.V. is supported by FWO PhD fellowship 11A0419N (https://www.fwo.be/en/fellowships-funding/phd-fellowships/). J.d.W. is supported by European Research Council (ERC) Starting Grant (#311083) (https://erc.europa.eu/funding/starting-grants); FWO Odysseus Grant; FWO Project grants G094016N and G0C4518N, FWO EOS grant G0H2818N; a Methusalem grant of KU Leuven/Flemish Government, and ERA-NET NEURON SynPathy 2015 (https://www.neuron-eranet.eu). The funders had no role in study design, data collection and analysis, decision to publish, or preparation of the manuscript.

**Competing interests:** The authors have declared that no competing interests exist.

**Abbreviations:** AIS, axon initial segment; AMPAR, α-amino-3-hydroxy-5-methyl-4-isoxazole propionic-type receptor; AnkG, Ankyrin-G; AP, affinity purification; APP, amyloid precursor protein; BSA, bovine serum albumin; Caspr2, contactin-associated protein-like 2; Cre, Cre recombinase; DIV, days in vitro; DN, dominant negative; EE, early endosome; EEA1, early endosome antigen 1; EGFP, enhanced green fluorescent protein; EPSC, excitatory postsynaptic current; eEPSC, evoked EPSC; eIPSC, evoked inhibitory postsynaptic current; ER, endoplasmic reticulum; GluA2, glutamate ionotropic receptor AMPA type subunit 2; HA, hemagglutinin; HDR, homology directed repair; IgG, immunoglobulin G; KDEL, endoplasmic reticulum retention signal KDEL; KI, knock in; KO, knock out; LAR, leukocyte common antigen-related protein; LRRTM1, leucine-rich repeat transmembrane neuronal protein 1; LV, lentivirus; L1/NgCAM, neuron-glia cell adhesion molecule L1; MAP2, microtubule-associated protein 2; mEPSC, miniature excitatory postsynaptic current; mIPSC, miniature inhibitory postsynaptic current; MS, mass spectrometry; NGL-3, Netrin-G Ligand 3; Nlgn1, Neuroligin 1; Nrxn, neurexin; PDZ, PSD-95/discs large/zona occludens-1 motif; Pves, vesicular release probability; Rab, Rab GTPase; Rab11FIP5, Rab11 family-interacting protein 5; RE, recycling endosome; Rip11, Rab11 interacting protein; RNP, ribonucleoprotein; RRP, readily releasable pool; RUSH, retention using selective hooks; SBP, streptavidin-binding protein; shRNA, short hairpin

neuropsychiatric disorders, especially schizophrenia and autism [10–12], underscoring the importance of Nrxns for circuit function. Despite their essential role, the molecular mechanisms that control presynaptic polarization and abundance of Nrxns are largely unknown [13]. Previous studies have shown that surface trafficking of Nrxns requires the C-terminal PSD-95/discs large/zona occludens-1 (PDZ)-binding motif [14,15] and that surface mobility depends on synaptic activity and interaction with extracellular ligands [16,17].

The vertebrate Sortilin-related CNS expressed 1–3 (SorCS1-3) proteins are sorting receptors belonging to the vacuolar protein sorting 10 (VPS10P) family receptors that have emerged as major regulators of intracellular protein trafficking [18,19]. SorCS proteins are required for the surface expression of glutamate receptors, neurotrophin receptors, transporters, and adhesion molecules [20–24]. These sorting receptors are thus essential for synaptic transmission and plasticity, but the mechanisms by which SorCS proteins sort their cargo are poorly understood. Using quantitative proteomics analysis, we recently established a critical role for SorCS1 in Nrxn surface trafficking [20]. Although our previous observations suggested that SorCS1 might regulate endosomal trafficking of Nrxn, the intracellular pathway via which Nrxn is trafficked to synapses, the regulation thereof by SorCS1, and the relevance of SorCS1-mediated sorting of Nrxn and other axonal cargo proteins for synaptic development and function remained elusive.

Here, we use epitope-tagging to elucidate the subcellular localization of endogenous SorCS1 and Nrxn1α. We show that SorCS1 acts in dendritic endosomes to control a balance between the axonal and dendritic surface distribution of Nrxn1α in the same neuron. Although predominantly considered to be a presynaptic molecule, we find that newly synthesized Nrxn1α first traffics to dendrites, where it is endocytosed, followed by transcytosis to the axon. SorCS1 controls Nrxn axonal surface polarization by facilitating the transition of endocytosed Nrxn from early to recycling endosomes and does so by interacting with Rab11 family-interacting protein 5 (Rab11FIP5)/Rab11 interacting protein (Rip11), a novel SorCS1 cytoplasmic binding partner we identify. In the absence of SorCS1, Nrxn1α accumulates in early endosomes and mispolarizes to the dendritic surface, impairing presynaptic differentiation induced by the postsynaptic Nrxn ligand Neuroligin (Nlgn). This defect can be rescued by a Nrxn1α mutant that bypasses SorCS1-mediated sorting and transcytosis to polarize to the axonal surface, but not by wild type (WT) Nrxn1α. Finally, we show that SorCS1 is required for presynaptic function. Together, our observations indicate that SorCS1-mediated sorting in dendritic endosomes controls Nrxn axonal surface polarization required for proper synapse development and function.

## Results

### SorCS1 controls an axonal–dendritic balance in Nrxn1α surface polarization

We first determined the subcellular distribution of endogenous SorCS1, which has been difficult to assess because of a lack of suitable antibodies. We generated a SorCS1 knock-in (KI) mouse (*Sorcs1*[HA]), using CRISPR/Cas9 to insert a hemagglutinin (HA) epitope tag in the *Sorcs1* locus after the propeptide domain (S1A Fig). Correct insertion of the HA tag was confirmed by sequencing and western blot analysis (S1B Fig). HA immunodetection in cultured *Sorcs1*[HA] cortical neurons revealed prominent punctate endogenous SorCS1 immunoreactivity in soma and dendrites (Fig 1A), which was mimicked by exogenously expressed HA-SorCS1 in hippocampal neurons (Fig 1B and 1C). These results are in agreement with a previous immunohistochemical study showing a somatodendritic distribution of SorCS1 [25] and

ribonucleic acid; SorCS1, Sortilin-related CNS expressed 1; TfR, transferrin receptor; TGN, trans-Golgi network; TKD, triple knock down; VPS10P, vacuolar protein sorting 10; WT, wild type.

suggest that SorCS1-mediated sorting of cargo proteins, including Nrxn, occurs in this cellular compartment.

To begin to elucidate how SorCS1 regulates membrane trafficking of its cargo Nrxn, we electroporated cortical *Sorcs1*$^{flox/flox}$ mouse neurons with Cre recombinase (Cre) to remove SorCS1 (*Sorcs1* knock out [KO]). We transfected neurons with HA-Nrxn1α, one of the most abundant Nrxn isoforms in the brain [26], and live-labeled them with an HA antibody to detect surface Nrxn1α, followed by Ankyrin-G (AnkG) and microtubule-associated protein 2 (MAP2) immunodetection to label the axonal and dendritic compartment, respectively. Loss of SorCS1 decreased Nrxn1α axonal surface levels and concomitantly increased dendritic surface levels in the same cell, changing Nrxn1α surface polarization from axonal to dendritic (Fig 1D and 1E). Re-expression of WT SorCS1 (SorCS1$^{WT}$) in Cre-positive *Sorcs1*$^{flox/flox}$ neurons restored Nrxn1α axonal surface polarization (Fig 1D and 1E), indicating that the defect occurred cell autonomously. In contrast, rescue with an endocytosis-defective mutant [27] of SorCS1 (SorCS1$^{Y1132A}$), which remains on the plasma membrane (S1C Fig and S1D Fig), did not rescue Nrxn1α surface polarization (Fig 1D and 1E). To determine whether SorCS1 controls the subcellular distribution of endogenous Nrxn, we immunostained *Sorcs1*$^{flox/flox}$ neurons electroporated with Cre or enhanced green fluorescent protein (EGFP) with a pan-Nrxn antibody directed against the conserved cytoplasmic tail [28] (S1E Fig and S1F Fig), which specifically labels endogenous Nrxn (S1G Fig and S1H Fig). Similar to the observations with transfected HA-Nrxn1α in *Sorcs1* KO neurons, loss of SorCS1 reduced axonal intensity of endogenous Nrxn and decreased the axonal–dendritic ratio compared with control neurons (S1E Fig and S1F Fig). Conversely, overexpression of SorCS1$^{WT}$ in hippocampal neurons, which express very low levels of *Sorcs1* [29], increased Nrxn1α axonal surface polarization and concurrently decreased dendritic surface levels, whereas overexpression of SorCS1$^{Y1132A}$ did not alter Nrxn1α surface polarization (S1I Fig and S1J Fig). Together, these results indicate that somatodendritic SorCS1 controls an axonal–dendritic surface balance of Nrxn1α. Our findings suggest that SorCS1 is either required for endocytosis of Nrxn1α or acts in endosomes to bias Nrxn1α surface polarization toward the axon.

### Indirect axonal surface trafficking of Nrxn1α in mature neurons

Nrxns are considered to be predominantly presynaptic proteins, although several independent studies have reported the presence of a dendritic pool of Nrxns [20,30–33]. To unequivocally determine the subcellular localization of endogenous Nrxn, we generated a Nrxn1α KI mouse (*Nrxn1α*$^{HA}$), inserting an HA epitope tag in the *Nrxn1α* locus after the signal peptide (S2A Fig). Correct insertion of the HA tag was confirmed by restriction digest, sequencing, and western blot analysis (S2B Fig). We cultured *Nrxn1α*$^{HA}$ cortical neurons together with WT cortical neurons to reliably detect endogenous HA-Nrxn1α in the somatodendritic and axonal compartment (S2C Fig and S2D Fig). At 3 days in vitro (DIV), HA-Nrxn1α surface distribution was polarized towards the axon (Fig 2A and 2B). As neurons matured, surface HA-Nrxn1α axonal surface levels decreased, whereas dendritic surface levels increased (Fig 2A and 2B). Surface HA-Nrxn1α remained polarized towards the axon at all developmental time points analyzed (Fig 2A and 2B). A similarly polarized distribution of Nrxn was observed in WT neurons labeled with the pan-Nrxn antibody (S2E Fig and S2F Fig). Together, these results indicate that endogenous Nrxn1α is an axonally polarized surface protein that accumulates in dendrites as neurons mature.

To determine how Nrxn1α is trafficked in cortical neurons, we employed the retention using selective hooks (RUSH) system [34] to retain Nrxn1α in the endoplasmic reticulum (ER) and induced synchronous release and transport through the secretory pathway by

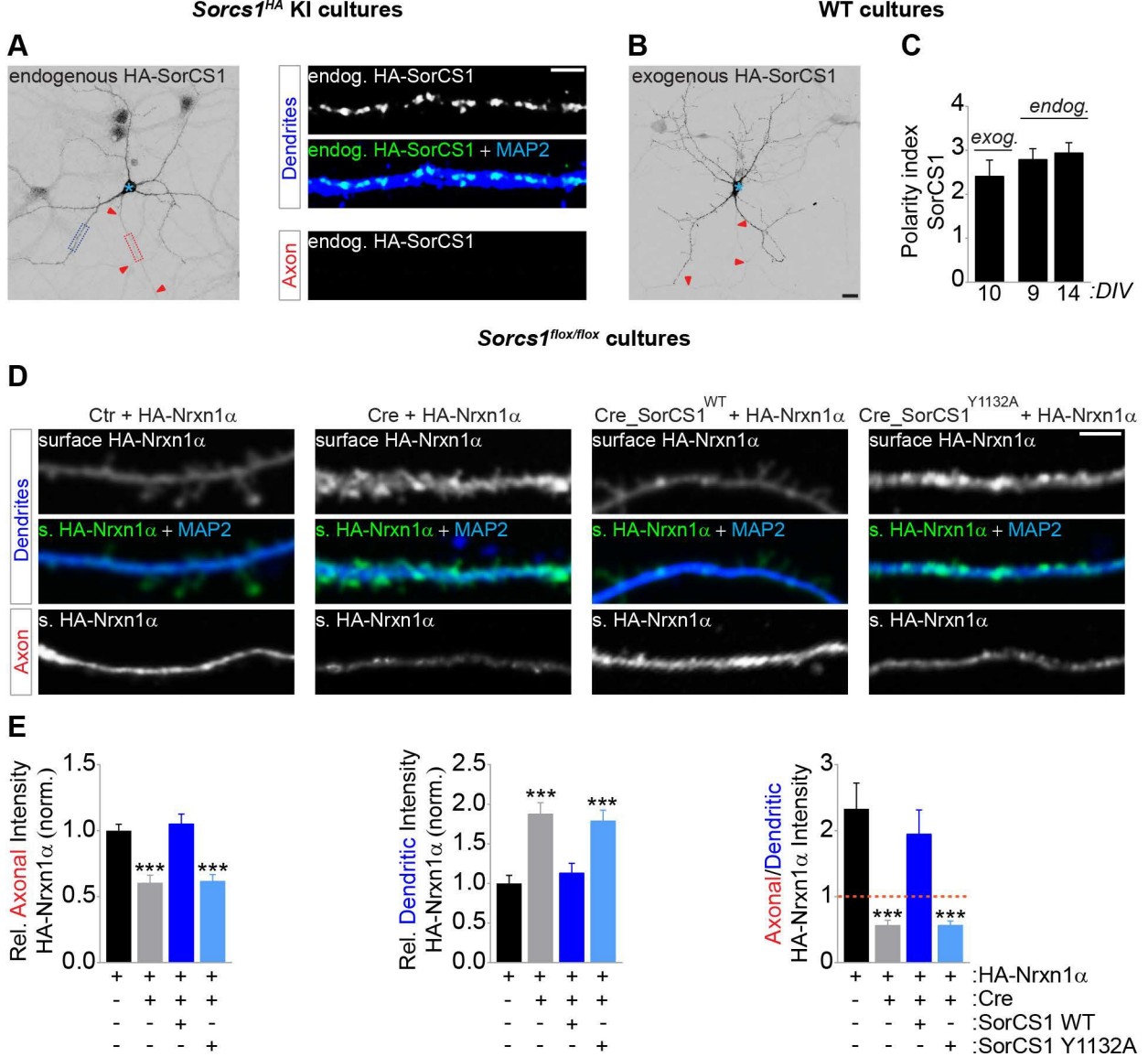

**Fig 1. SorCS1-mediated sorting controls an axonal–dendritic surface balance of Nrxn1α.** (A) DIV9 *Sorcs1^HA* mouse cortical neurons immunostained for endogenous (endog.) HA-SorCS1 (grayscale and green) and MAP2 (blue). Red arrowheads indicate the axon, and the blue asterisk marks the cell body. High-zoom images show dendritic (dotted blue box) and axonal (dotted red box) distribution of HA-SorCS1. (B) DIV10 WT mouse hippocampal neurons transfected with HA-SorCS1 and immunostained for HA (grayscale). (C) Quantification of panels A and B: dendritic versus axonal distribution (D:A–polarity index) of endogenous (DIV9, *n* = 26 neurons and DIV14, *n* = 26) and exogenous (exog.; *n* = 14) HA-SorCS1 from 3 and 2 independent experiments, respectively. (D) DIV8 to DIV10 *Sorcs1^flox/flox* mouse cortical neurons electroporated with EGFP (Ctr), Cre-EGFP, Cre-EGFP-T2A-SorCS1^WT, or Cre-EGFP-T2A-SorCS1^Y1132A and transfected with HA-Nrxn1α, immunostained for surface (s.) HA-Nrxn1α (grayscale and green) and MAP2 (blue). (E) Quantification of panel D: surface HA-Nrxn1α fluorescence intensity in axon and dendrites relative to total surface levels and normalized to cells expressing EGFP, and ratio of axonal–dendritic surface HA intensity. Ctr (*n* = 27 neurons); Cre (*n* = 29); Cre_SorCS1^WT (*n* = 28); Cre_SorCS1^Y1132A (*n* = 28). ***$P < 0.001$ (Kruskal-Wallis test followed by Dunn's multiple comparisons test, 3 independent experiments). Underlying numerical values can be found in S1 Data. Graphs show mean ± SEM. Scale bars, 20 μm (panel B); 5 μm (panel A [high-zoom], panel D). Cre, Cre recombinase; Ctr, control; DIV, days in vitro; EGFP, enhanced green fluorescent protein; HA, hemagglutinin; MAP2, microtubule-associated protein 2; Nrxn, neurexin; SorCS1, Sortilin-related CNS expressed 1; WT, wild type.

applying biotin (Fig 2C). We performed live-cell imaging in 8 through 10 DIV rat cortical neurons co-expressing streptavidin-binding protein (SBP)-EGFP-Nrxn1α (reporter) and streptavidin-ER retention signal KDEL (KDEL) (ER hook), which were live-labeled with an antibody

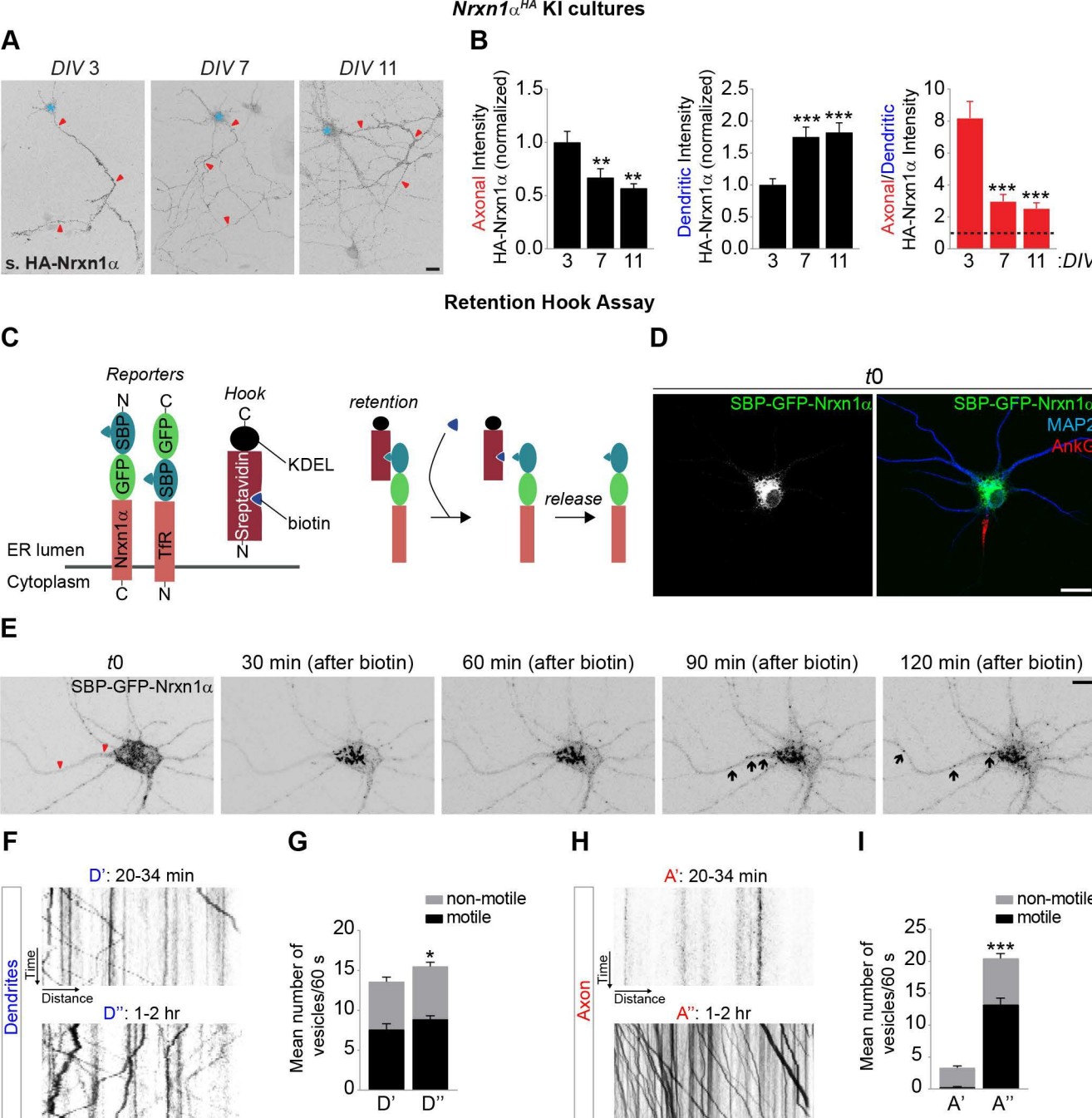

**Fig 2. Indirect axonal trafficking of Nrxn1α in mature neurons.** (A) Nonpermeabilized DIV3, DIV7, and DIV11 *Nrxn1α*[HA] cortical neurons live-labeled with an HA antibody to visualize endogenous surface (s.) HA-Nrxn1α (grayscale) in axon and dendrites. Red arrowheads indicate the axon, and the blue asterisk marks the cell body. (B) Quantification of panel A: surface HA-Nrxn1α fluorescence intensity in axon and dendrites normalized to DIV3 neurons and the ratio of axonal–dendritic surface HA-Nrxn1α intensity. DIV3 ($n = 30$ neurons); DIV7 ($n = 29$); DIV11 ($n = 24$). $**P < 0.01$; $***P < 0.001$ (Kruskal-Wallis test followed by Dunn's multiple comparisons test, 3 independent cultures). (C) Schematic representation of streptavidin-KDEL (ER hook) and reporters (Nrxn1α and TfR) used in RUSH experiments. Addition of biotin dissociates the reporters from the ER hook, inducing synchronous release from the ER and transport through the secretory pathway. (D) DIV9 WT mouse cortical neuron co-expressing SBP-EGFP-Nrxn1α and streptavidin-KDEL immunostained for MAP2 (blue), Ankyrin-G (red), and EGFP-Nrxn1α (grayscale and green) at t = 0 before adding biotin. (E) Live-cell imaging in DIV8 to DIV10 WT rat cortical neurons co-expressing SBP-EGFP-Nrxn1α and ER hook. After 24 to 31 h of expression, neurons were imaged every 5 min for 2.5 h. Biotin was added 10 min after the beginning of the imaging session. Shown are representative images of SBP-EGFP-Nrxn1α fluorescence in dendrites and axon before (t0) and 30, 60, 90 and 120 min after adding biotin. Red arrowheads indicate axon and black arrows indicate SBP-EGFP-Nrxn1α-positive puncta. See also S1 Movie. (F–I) Live-cell imaging in DIV8 to DIV10 WT rat cortical neurons co-expressing SBP-EGFP-Nrxn1α and ER hook. After 21 to 30

h of expression, neurons were imaged every second for 60 to 120 s either 20 to 34 min or 1 to 2 h after adding biotin. See also S2 Movie and S3 Movie. (F) Kymographs illustrating EGFP-Nrxn1α vesicle dynamics over a 60 s period in dendrites (D) from neurons treated with biotin for either 25 min or 2 h. (G) Mean number of motile and nonmotile EGFP-Nrxn1α vesicles in dendrites from neurons treated with biotin for either 20 to 34 min ($n = 17$ neurons) or 1 to 2 h ($n = 16$) in 2 and 3 independent experiments, respectively. $^*P < 0.05$ (Mann-Whitney test). (H) Kymographs illustrating EGFP-Nrxn1α vesicle dynamics over a 60 s period in axons (A) from neurons treated with biotin for either 25 min or 2 h. (I) Mean number of motile and nonmotile EGFP-Nrxn1α vesicles in axons. $^{***}P < 0.001$ (Mann-Whitney test). Underlying numerical values can be found in S1 Data. Graphs show mean ± SEM. Scale bars, 20 μm (panels A and D); 10 μm (panel E). AnkG, Ankyrin-G; DIV, days in vitro; ER, endoplasmic reticulum; EGFP, enhanced green fluorescent protein; HA, hemagglutinin; KDEL, endoplasmic reticulum retention signal KDEL; KI, knock in; MAP2, microtubule-associated protein 2; Nrxn, neurexin; RUSH, retention using selective hooks; SBP, streptavidin-binding protein; TfR, transferrin receptor; WT, wild type.

against the axon initial segment (AIS) protein Neurofascin to visualize the axonal compartment (S3A Fig). At t = 0, we observed diffuse SBP-EGFP-Nrxn1α fluorescence in soma and dendrites, in a pattern resembling the ER (Fig 2D). Following application of biotin, Nrxn1α fluorescence coalesced into punctate structures that started trafficking abundantly in the somatodendritic compartment, followed by Nrxn1α trafficking in the axon after a marked delay (Fig 2E, S1 Movie). Quantification of Nrxn1α fluorescence intensity showed a decrease in Nrxn1α intensity in the soma and an increase in axonal intensity approximately 90 min after biotin application (S3B Fig). Kymograph analysis showed a small increase in the mean number of motile vesicles present in dendrites at late (1–2 h) compared with early (20–34 min) time points (Fig 2F and 2G, S2 Movie, S3 Movie). Shortly after ER release, the majority of vesicles in dendrites moved in the anterograde direction, shifting to a retrograde movement at the later time point (S3C Fig). Strikingly, motile Nrxn1α vesicles were absent in axons shortly after ER release but increased dramatically at late time points (Fig 2H and 2I, S2 Movie, S3 Movie). The vast majority of these vesicles moved anterogradely (S3D Fig).

We performed several controls to verify that the delayed axonal trafficking of Nrxn1α is not an artefact of prolonged retention in the ER and synchronized release, which might overwhelm the trafficking machinery. Trafficking of the somatodendritic cargo protein transferrin receptor (TfR) [35] was detected only in the somatodendritic compartment (S3E Fig and S3F Fig, S4 Movie). In the absence of biotin, dendritic SBP-EGFP-Nrxn1α-positive structures were not motile (S5 Movie), indicating that the delayed trafficking of Nrxn1α to axons is not simply caused by prolonged ER retention. Moreover, immature DIV3 neurons displayed Nrxn1α trafficking to the axonal compartment shortly after biotin application (S3G Fig and S3H Fig, S6 Movie), further indicating that the delayed axonal trafficking of Nrxn1α in mature neurons does not result from prolonged retention in the ER but correlates with the maturation state of the neuron. Taken together, these results show that following transport through the secretory pathway, Nrxn1α is first trafficked to the somatodendritic compartment and appears in axons with a marked delay.

## Nrxn1α is transcytosed from the dendritic surface to axons

Our observations suggest a dendrite-to-axon transcytotic pathway for axonal surface polarization of Nrxn1α. The transcytotic model predicts that Nrxn1α is first trafficked to the somatodendritic plasma membrane, where it is internalized into endosomes and trafficked to the axonal compartment via endosome-derived carriers [36]. To determine whether Nrxn1α is internalized from the dendritic plasma membrane, we followed the fate of Nrxn1α throughout the endosomal pathway. Internalised Nrxn1α was largely restricted to the dendritic compartment (S4A Fig and S4B Fig) and shifted from early endosomes (EEs; early endosome antigen 1 [EEA1]- and Rab5-positive) to recycling endosomes (REs; Rab11-positive) over time (S4C–S4I Fig). Blocking of dynamin-dependent endocytosis, either by expressing a dominant-negative (DN) mutant of Dynamin1 (K44A) or with Dynasore (S5A–S5D Fig), caused a decrease in

Nrxn1α axonal surface levels and a concomitant increase in dendritic surface levels. Expression of GTP binding-deficient mutants of Rab5 (S34N) and Rab11 (S25N) to interfere with transport to EEs and REs, respectively, mimicked the effect of blocking endocytosis (S5E–S5H Fig). However, expression of Rab7 (T22N) to interfere with transport to late endosomes did not affect the axonal surface polarization of Nrxn1α (S5I Fig and S5J Fig). Thus, endocytosis, EE and RE transport, but not late endosomal transport, are required for accumulation of Nrxn1α on the axonal plasma membrane, indicating transcytotic trafficking of Nrxn1α.

## A SorCS1-Rab11FIP5 interaction controls Nrxn1α transition from early to REs

The transcytotic trafficking route of Nrxn1α and somatodendritic distribution of SorCS1 suggest that sorting of Nrxn1α occurs in dendrites. To determine how SorCS1 controls transcytotic trafficking of Nrxn1α, we followed the fate of internalized HA-Nrxn1α in *Sorcs1* KO neurons. Endocytosis of HA-Nrxn1α in dendrites was not affected in *Sorcs1* KO neurons compared with control cells (S6A Fig and S6B Fig). We next analyzed colocalization of internalized Nrxn1α with EEs (EAA1), fast REs (Rab4), REs (Rab11) in dendrites, and REs (Rab11) in the axon, respectively, of *Sorcs1* KO neurons. Quantification revealed an increase in the colocalization of internalized Nrxn1α with EAA1-positive EEs and Rab4-positive fast REs in dendrites of *Sorcs1* KO neurons (Fig 3A–3D). Colocalization of internalized Nrxn1α with Rab11-positive REs in *Sorcs1* KO neurons was decreased in dendrites (Fig 3E and 3F) and even more strongly reduced in the axon (Fig 3G and 3H). Thus, endocytosed Nrxn1α accumulates in EEs and is missorted to Rab4-fast REs in the absence of SorCS1-mediated sorting. These observations are consistent with our finding that dendritic surface levels of Nrxn1α are increased in *Sorcs1* KO neurons (Fig 1D and 1E), which is likely due to increased recycling of Nrxn1α via Rab4-fast REs back to the dendritic plasma membrane. Similarly, the reduced axonal surface levels of Nrxn1α in *Sorcs1* KO neurons (Fig 1D and 1E) likely result from decreased sorting of Nrxn1α to Rab11-REs. Together, these results demonstrate that SorCS1 controls axonal surface polarization of Nrxn1α by facilitating the transition from EEs to Rab11-REs.

We reasoned that SorCS1 might interact with additional proteins to facilitate Nrxn1α sorting from early to REs. Rab11FIP5 or Rip11 is prominently present in the raw mass spectrometry (MS) data set we obtained after affinity purification (AP) of SorCS1 complexes from rat brain extracts with 2 independent antibodies [20] (AP-MS data available online). Rip11, which belongs to the family of Rab11-interacting proteins [37], localizes to REs and regulates transcytosis of proteins from the basolateral to the apical plasma membrane in polarized epithelial cells [38,39]. Western blot analysis of immunoprecipitated HA-SorCS1 from postnatal *Sorcs1*^HA KI cortical extracts showed a robust Rip11 band coprecipitating with SorCS1 but not with control mouse immunoglobulin G (IgG) (Fig 3I). Rip11 displayed a punctate distribution in dendrites and colocalized with SorCS1 (Fig 3J and 3K). Expression of a DN form of Rip11 that inhibits the transport from early to REs [40] reduced axonal surface levels of HA-Nrxn1α and increased dendritic surface levels, shifting Nrxn1α surface polarization from axonal to dendritic (Fig 3L and 3M), similar to loss of SorCS1. Expression of WT Rip11, on the other hand, increased axonal surface polarization of HA-Nrxn1α (Fig 3L and 3M). Together, these results demonstrate that a SorCS1-Rip11 interaction facilitates Nrxn1α sorting from EEs to REs, preventing missorting to fast REs and late endosomes and biasing trafficking to the axonal surface.

## Selective missorting of transcytotic cargo in the absence of SorCS1

We next asked whether the surface polarization of other axonal membrane proteins was affected in the absence of SorCS1. The cell adhesion molecules neuron-glia cell adhesion

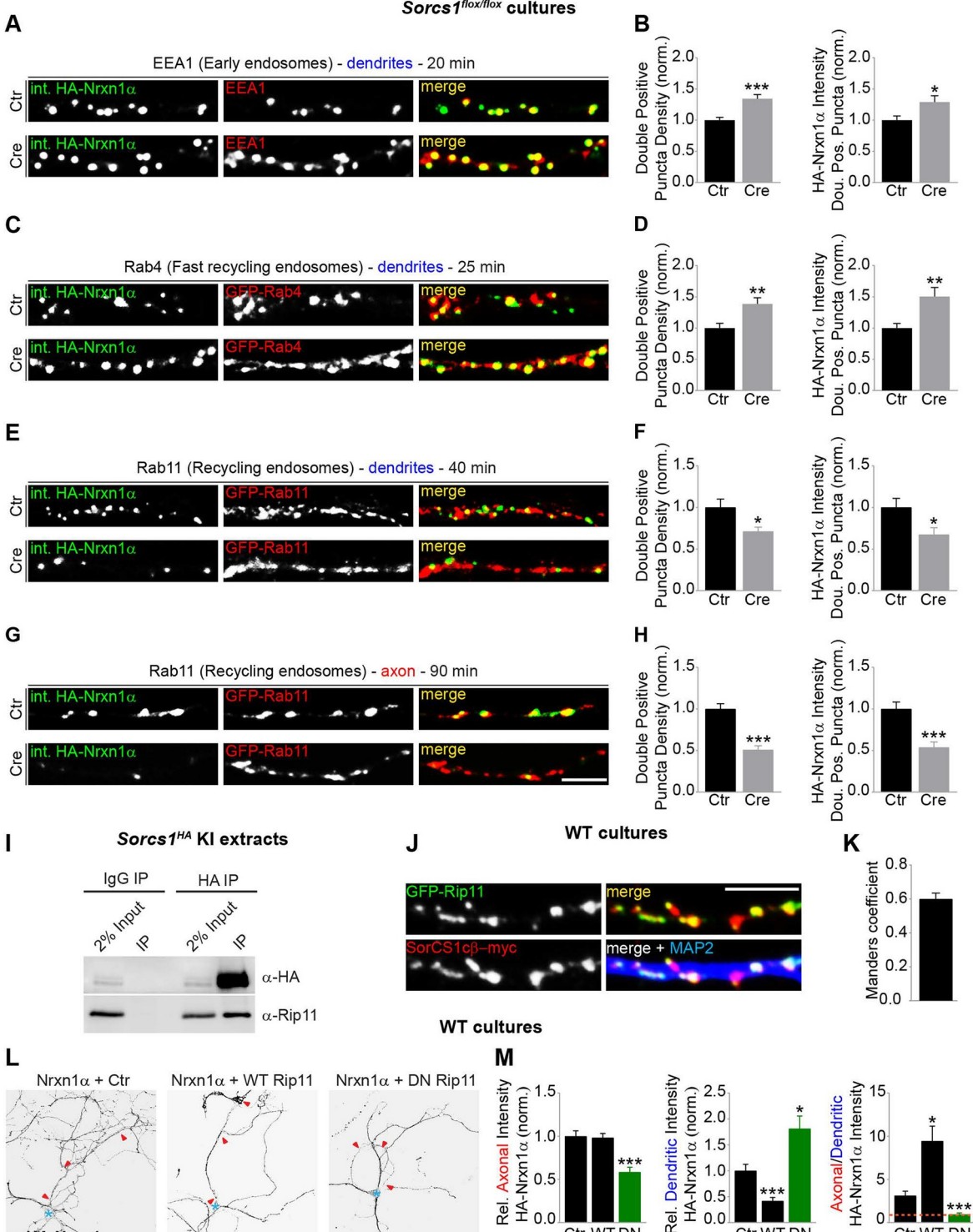

**Fig 3. SorCS1 interacts with Rab11FIP5/Rip11 to regulate Nrxn1α transition from EEs to REs.** (A) DIV8 to DIV10 *Sorcs1^flox/flox* cortical neurons electroporated with mCherry (Ctr) or Cre-T2A-mCherry and transfected with HA-Nrxn1α (pulse-chased for 20 min), labeled for internalized (int.) HA-Nrxn1α (grayscale and green) and EEA1 (grayscale and red). (B) Quantification of panel A: number of internalized Nrxn1α- and EEA1-double-positive puncta normalized to cells expressing mCherry and intensity of internalized Nrxn1α fluorescence in the double-positive puncta normalized to cells expressing mCherry. Ctr (*n* = 26 neurons); Cre (*n* = 25). *P < 0.05; ***P < 0.001 (Mann-Whitney test, 3 independent experiments). (C) DIV8 to DIV10 *Sorcs1^flox/flox* cortical neurons electroporated with mCherry (Ctr) or Cre-T2A-mCherry and co-transfected with HA-Nrxn1α and EGFP-Rab4 (pulse-chased for 25 min), labeled for

internalized HA-Nrxn1α (grayscale and green) and EGFP-Rab4 (grayscale and red). (D) Quantification of panel C; Ctr (*n* = 28 neurons); Cre (*n* = 26). **P < 0.01 (Mann-Whitney test, 3 independent experiments). (E) DIV8 to DIV10 *Sorcs1^{flox/flox}* cortical neurons electroporated with mCherry (Ctr) or Cre-T2A-mCherry and co-transfected with HA-Nrxn1α and EGFP-Rab11 (pulse-chased for 40 min), labeled for internalized HA-Nrxn1α (grayscale and green) and EGFP-Rab11 (grayscale and red). (F) Quantification of panel E; Ctr (*n* = 29 neurons); Cre (*n* = 23). *P < 0.05 (Mann-Whitney test, 3 independent experiments). (G) DIV8 to DIV10 *Sorcs1^{flox/flox}* cortical neurons electroporated with mCherry (Ctr) or Cre-T2A-mCherry and co-transfected with HA-Nrxn1α and EGFP-Rab11 (pulse-chased for 90 min), labeled for internalized HA-Nrxn1α (grayscale and green) and EGFP-Rab11 (grayscale and red). (H) Quantification of panel G; Ctr (*n* = 28 neurons); Cre (*n* = 26). ***P < 0.001 (Mann-Whitney test, 3 independent experiments). (I) Western blot for the recovery of Rab11FIP5/Rip11 in immunoprecipitated HA-SorCS1 complexes from P21–P28 *Sorcs1^{HA}* cortical prey extracts. See S9 Fig for raw uncropped blots. (J) DIV9 to DIV10 WT cortical neurons co-expressing EGFP-Rip11 and SorCS1cβ-myc immunostained for EGFP-Rip11 (grayscale and green), SorCS1-myc (grayscale and red) and MAP2 (blue). (K) Quantification of the colocalization of Rip11 with SorCS1 expressed as Manders coefficient (*n* = 20 neurons) in 2 independent experiments. (L) DIV9 WT cortical neurons co-expressing HA-Nrxn1α and EGFP (Ctr), WT EGFP-Rip11 or DN EGFP-Rip11, and immunostained for surface (s.) HA-Nrxn1α (grayscale). Red arrowheads indicate the axon, and the blue asterisk marks the cell body. (M) Quantification of panel L: surface HA-Nrxn1α fluorescence intensity in axon and dendrites relative to total surface levels and normalized to cells expressing EGFP and ratio of axonal–dendritic surface HA intensity (*n* = 30 neurons for each group). *P < 0.05; ***P < 0.001 (Kruskal-Wallis test followed by Dunn's multiple comparisons test, 3 independent experiments). Underlying numerical values can be found in S1 Data. Graphs show mean ± SEM. Scale bars, 5 μm (panels G and J); 20 μm (panel L). Cre, Cre recombinase; Ctr, control; DIV, days in vitro; DN, dominant negative; EE, early endosome; EEA1, early endosome antigen 1; EGFP, enhanced green fluorescent protein; HA, hemagglutinin; MAP2, microtubule-associated protein 2; Nrxn, neurexin; Rab, Rab GTPase; RE, recycling endosome; Rip11, Rab11 interacting protein; SorCS1, Sortilin-related CNS expressed 1; WT, wild type.

molecule L1 (L1/NgCAM) and contactin-associated protein-like 2 (Caspr2) are both targeted to the axonal surface but via distinct trafficking routes. L1/NgCAM is transcytosed from dendrites to axon [41,42], whereas Caspr2 undergoes nonpolarized delivery followed by selective endocytosis from the somatodendritic surface resulting in axonal polarization [43]. L1/NgCAM levels are down-regulated in *Sorcs1* KO synaptosomes [20]. *Sorcs1^{flox/flox}* cortical neurons electroporated with Cre or EGFP were transfected with myc-L1 or HA-Caspr2 and immunostained for surface myc or HA. Consistent with a role for SorCS1 in regulating transcytosed axonal cargo, surface polarization of L1/NgCAM, but not of Caspr2, was perturbed in *Sorcs1* KO neurons (S6C–S6E Fig). The polarity index of 2 somatodendritic proteins (glutamate ionotropic receptor AMPA type subunit 2 [GluA2-EGFP] and MAP2) was unchanged in *Sorcs1* KO neurons (S7 Fig), indicating that loss of SorCS1 does not generally perturb the polarized distribution of neuronal proteins. Thus, loss of SorCS1 selectively impairs transcytotic trafficking of axonal membrane proteins, while keeping other neuronal polarity mechanisms intact.

## Loss of SorCS1 impairs Nrxn-mediated presynaptic differentiation

We next sought to determine motifs in the Nrxn1α protein sequence that contain signals necessary for its membrane trafficking, with the goal of identifying a Nrxn1α mutant that can bypass SorCS1-mediated sorting and transcytotic trafficking. We systematically tested the effect of a series of Nrxn1α cytoplasmic deletions on surface polarization (Fig 4A and 4B, S8A–S8C Fig) and found that removing the 4.1-binding motif in the cytoplasmic domain (Nrxn1α Δ4.1) was the only deletion that dramatically increased Nrxn1α axonal surface polarization (Fig 4A and 4B, S8A–S8C Fig). Remarkably, axonal surface polarization of Nrxn1α Δ4.1 was unaffected by loss of SorCS1 (Fig 4C and 4D). Moreover, Dynasore treatment to block endocytosis did not impair axonal surface polarization of Nrxn1α Δ4.1 (Fig 4E and 4F), indicating that the Nrxn1α Δ4.1 mutant bypasses dendritic endocytosis and sorting by SorCS1 to polarize to the axonal surface.

Nrxn missorting to the dendritic surface and its concomitant loss from the axonal surface in *Sorcs1* KO neurons would be expected to impair synapse formation on heterologous cells expressing a postsynaptic ligand for Nrxn [44]. To test this, we infected *Sorcs1^{flox/flox}* cortical neurons with lentivirus (LV) to express Cre-T2A-mCherry or mCherry as control and co-

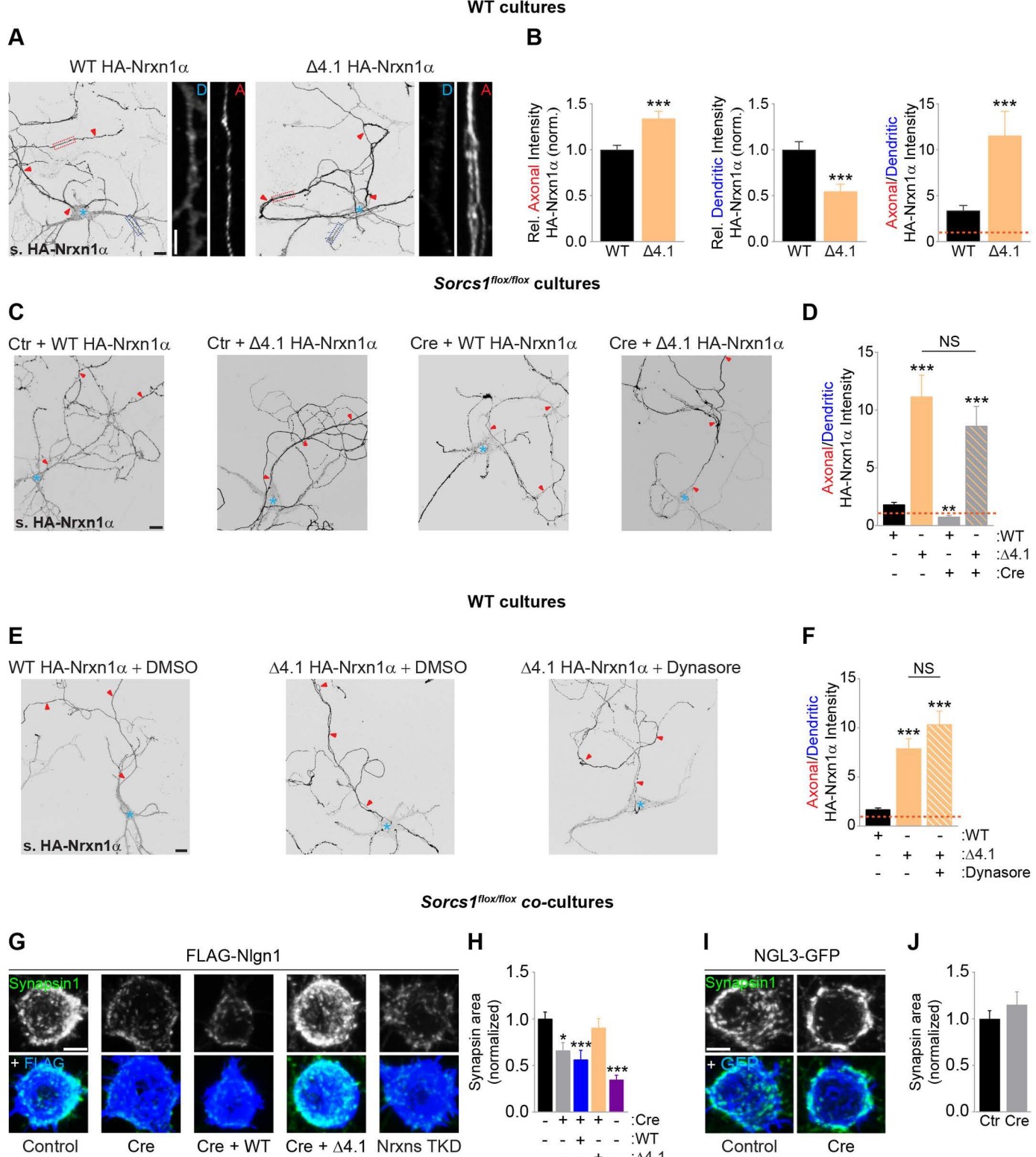

**Fig 4. SorCS1-mediated axonal surface polarization of Nrxn is required for presynaptic differentiation.** (A) DIV8 to DIV10 WT mouse cortical neurons transfected with WT HA-Nrxn1α and a cytoplasmic deletion mutant lacking the 4.1-binding motif (HA-Nrxn1α Δ4.1) and immunostained for surface (s.) HA-Nrxn1α (grayscale). Red arrowheads indicate the axon, and the blue asterisk marks the cell body. High-zoom images show dendritic (D, dotted blue box) and

axonal (A, dotted red box) HA-Nrxn1α. (B) Quantification of panel A: surface HA-Nrxn1α fluorescence intensity in axon and dendrites relative to total surface levels and normalized to cells expressing WT-Nrxn1α and ratio of axonal–dendritic surface HA intensity. WT ($n = 30$ neurons); Δ4.1 ($n = 29$). $^{***}P < 0.001$ (Mann-Whitney test, 3 independent experiments). (C) DIV8 to DIV10 mouse $Sorcs1^{flox/flox}$ cortical neurons electroporated with EGFP (Ctr) or Cre-EGFP (Cre) and transfected with WT HA-Nrxn1α or HA-Nrxn1α Δ4.1. Neurons were immunostained for surface HA-Nrxn1α (grayscale). (D) Quantification of panel C: ratio of axonal–dendritic surface HA-Nrxn1α fluorescence intensity. Ctr_WT ($n = 46$ neurons); Ctr_Δ4.1 ($n = 25$); Cre_WT ($n = 30$); Cre_Δ4.1 ($n = 27$). $^{**}P < 0.01$; $^{***}P < 0.001$ (Kruskal-Wallis test followed by Dunn's multiple comparisons test, at least 3 independent experiments). (E) DIV8 to DIV10 WT mouse cortical neurons transfected with WT HA-Nrxn1α or HA-Nrxn1α Δ4.1 and treated with DMSO (vehicle) or Dynasore. Neurons were immunostained 18 h after treatment for surface HA-Nrxn1α (grayscale). (F) Quantification of panel E: ratio of axonal–dendritic surface HA fluorescence intensity. WT_DMSO ($n = 40$ neurons); Δ4.1_DMSO ($n = 40$); Δ4.1_Dynasore ($n = 28$). $^{***}P < 0.001$ (Kruskal-Wallis test followed by Dunn's multiple comparisons test, at least 3 independent experiments). (G) HEK293T-cells expressing FLAG-Nlgn1 co-cultured with DIV10 $Sorcs1^{flox/flox}$ cortical neurons infected with LV expressing mCherry (Control), Cre-T2A-mCherry, Cre-T2A-mCherry-T2A-WT Nrxn1α, Cre-T2A-mCherry-T2A-Nrxn1α Δ4.1, or Nrxn TKD; immunostained for FLAG (blue) and Synapsin1 (grayscale and green). (H) Quantification of panel G: the area of Synapsin1 clustering on the surface of Nlgn1-expressing HEK cells and normalized to cells expressing mCherry. Control ($n = 29$ neurons); Cre ($n = 30$); Cre-WT ($n = 30$); Cre-Δ4.1 ($n = 30$); TKD Nrxns ($n = 30$). $^{*}P < 0.05$; $^{***}P < 0.001$ (Kruskal-Wallis test followed by Dunn's multiple comparisons test, 3 independent experiments). (I) HEK293T-cells expressing NGL-3-EGFP co-cultured with DIV10 $Sorcs1^{flox/flox}$ cortical neurons infected with LV expressing mCherry (Control) or Cre-T2A-mCherry and immunostained for EGFP (blue) and Synapsin1 (grayscale and green). (J) Quantification of panel I. Control ($n = 30$ neurons); Cre ($n = 30$). Underlying numerical values can be found in S1 Data. Graphs show mean ± SEM. Scale bars, 20 μm (panels A, C, and E); 5 μm (panels A [high-zoom], G, and I). Cre, Cre recombinase; Ctr, control; DIV, days in vitro; EGFP, enhanced green fluorescent protein; HA, hemagglutinin; HEK293T, human embryonic kidney cells; LV, lentivirus; NGL-3, Netrin-G Ligand 3; Nrxn, neurexin; NS, nonsignificant; SorCS1, Sortilin-related CNS expressed 1; TKD, triple knock down; WT, wild type.

cultured these with HEK293T cells expressing the Nrxn ligand FLAG-Neuroligin 1 (Nlgn1). Nlgn1-induced clustering of the presynaptic marker Synapsin 1 was reduced in $Sorcs1$ KO axons compared with control axons (Fig 4G and 4H). This defect was specific, because presynaptic differentiation induced by Netrin-G Ligand 3 (NGL-3), which requires presynaptic leukocyte common antigen-related protein (LAR) [45], was not affected in $Sorcs1$ KO neurons (Fig 4I and 4J). Infection with a lentiviral vector expressing short hairpin ribonucleic acids (shRNAs) against all Nrxns (Nrxn triple knock down [TKD]) [46] mimicked the defect in $Sorcs1$ KO neurons (Fig 4G and 4H), suggesting that Nlgn1-induced presynaptic differentiation is impaired in $Sorcs1$ KO neurons because of decreased axonal surface levels of Nrxn. To test this, we overexpressed HA-Nrxn1α in $Sorcs1$ KO neurons using LV. As expected, WT Nrxn1α did not rescue the impaired Nlgn1-mediated synaptogenic activity (Fig 4G and 4H) because of Nrxn1α mispolarization to the dendritic surface in $Sorcs1$ KO neurons. In contrast, expression of the Nrxn1α Δ4.1 mutant, which bypasses SorCS1-mediated sorting and the transcytotic route to polarize to the axonal surface, rescued the synaptogenic defect caused by SorCS1 loss (Fig 4G and 4H). Together, these results show that SorCS1-mediated sorting of Nrxns, by promoting Nrxn accumulation on the axonal surface, is required for normal synaptogenesis onto Nlgn1-expressing cells.

## SorCS1-mediated sorting is required for presynaptic function

Missorting of Nrxns, which regulate neurotransmitter release [5–7], would also be expected to impair presynaptic function. To assess the consequences of loss of SorCS1 on synaptic function, we recorded spontaneous miniature excitatory postsynaptic currents (mEPSCs) from $Sorcs1^{flox/flox}$ autaptic cortical cultures electroporated with Cre or EGFP. SorCS1 loss strongly decreased mEPSC frequency (Fig 5A and 5B). Decay kinetics and amplitude of mEPSCs were not altered by loss of SorCS1 (Fig 5A and 5B), suggesting that the amount of neurotransmitter release per vesicle and postsynaptic receptor properties and/or receptor density were not affected. The amplitude and total charge transfer of single evoked EPSCs (eEPSCs) were reduced in $Sorcs1$ KO neurons (Fig 5C and 5D). The mEPSC frequency and eEPSC amplitude defects can be attributed to a decrease in the readily releasable pool (RRP) size, in the vesicular release probability (Pves), or in active synapse number. Indeed, we previously found an increase in the number of silent excitatory synapses after loss of SorCS1 [20]. To assess RRP size, we applied a hyperosmotic sucrose stimulus. The amplitude and total charge transfer of

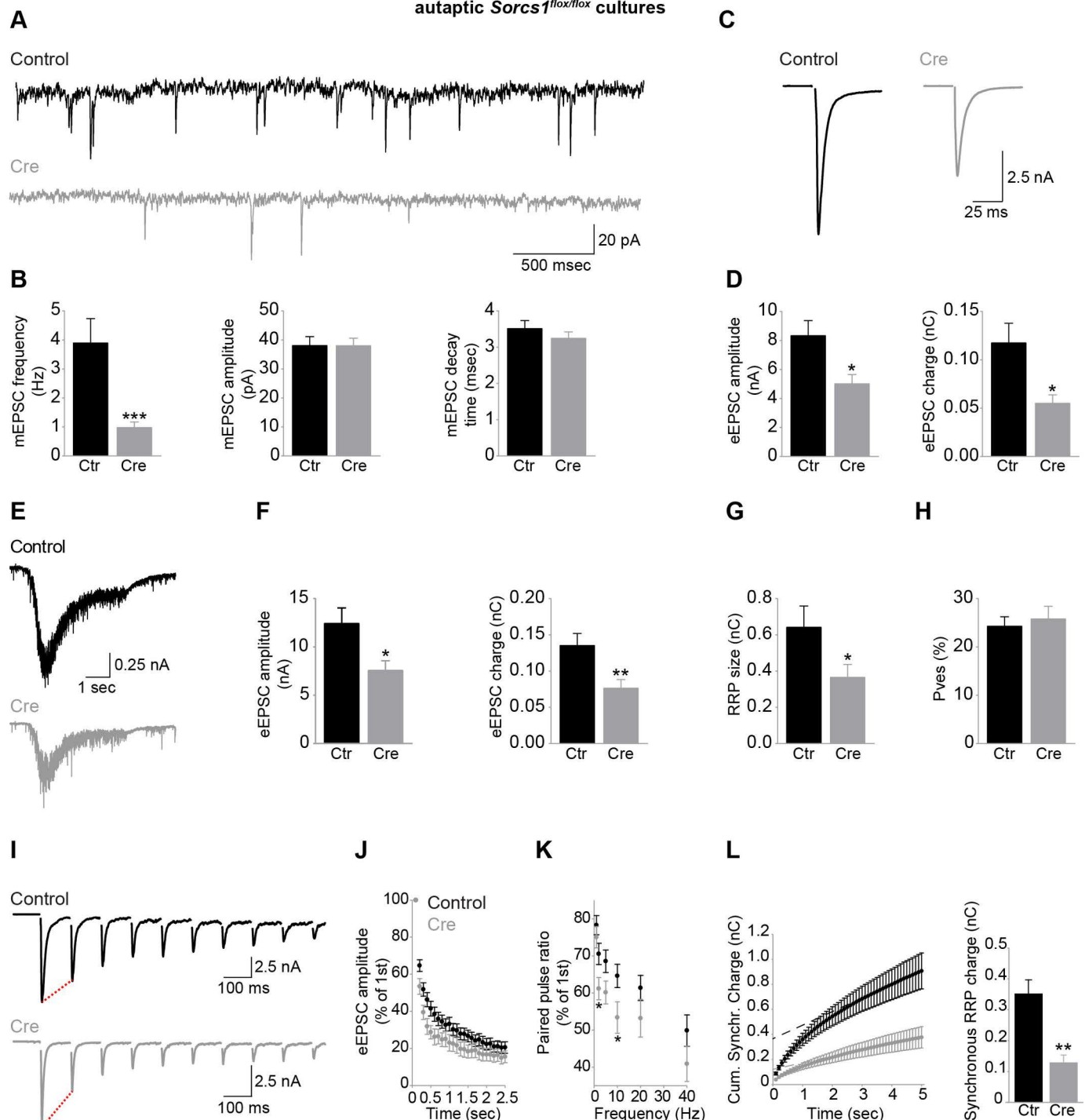

**Fig 5. SorCS1 is required for presynaptic function.** (A) Example traces of mEPSCs recorded from DIV14 to DIV16 *Sorcs1flox/flox* autaptic cortical neurons electroporated with EGFP (Control) or Cre-EGFP (Cre). (B) mEPSC frequency, but not amplitude and decay time, is decreased in *Sorcs1* KO neurons. Control (*n* = 18 neurons); Cre (*n* = 15). ***$P < 0.001$ (Mann-Whitney test, 3 independent experiments). (C) Example traces of eEPSCs recorded from DIV14 to DIV16 *Sorcs1flox/flox* autaptic cortical neurons electroporated with EGFP or Cre-EGFP. (D) eEPSC amplitude and eEPSC charge are decreased in *Sorcs1* KO neurons. Control (*n* = 17 neurons); Cre (*n* = 14). *$P < 0.05$ (Mann-Whitney test, 3 independent experiments). (E) Example traces of sucrose responses recorded from DIV14 *Sorcs1flox/flox* autaptic cortical neurons electroporated with EGFP or Cre-EGFP. (F–H) Decreased eEPSC amplitude, eEPSC charge (F) and RRP size (G) in *Sorcs1* KO neurons, but unaltered Pves (H). Control (*n* = 17 neurons); Cre (*n* = 21). *$P < 0.05$; **$P < 0.01$ (Mann-Whitney test, 3 independent experiments). (I) Example traces of train stimulations (10 Hz) recorded from DIV14 to DIV16 *Sorcs1flox/flox* autaptic cortical neurons electroporated with EGFP or Cre. (J) Increased depression during 10 Hz stimulation in *Sorcs1* KO neurons. Control (*n* = 18 neurons); Cre (*n* = 13); 3 independent experiments. (K) Increased paired-pulse

depression in *Sorcs1* KO neurons throughout different interstimulation intervals. Control ($n$ = 18 neurons); Cre ($n$ = 14). $^*P$ < 0.05 (two-tailed $t$ test, 3 independent experiments). (L) Estimation of the RRP size used during neuronal activity by back-extrapolating a linear fit of the steady state current towards the ordinate axis intercept, which represents the initial RRP size before train stimulations (10 Hz). RRP size of active synapses was reduced in DIV14 to DIV16 *Sorcs1* KO neurons. Control ($n$ = 18 neurons); Cre ($n$ = 13). $^{**}P$ < 0.01 (Mann-Whitney test, 3 independent experiments). Underlying numerical values can be found in S1 Data. Graphs show mean ± SEM. Cre, Cre recombinase; DIV, days in vitro; eEPSC, evoked excitatory postsynaptic current; EGFP, enhanced green fluorescent protein; KO, knock out; mEPSC, miniature excitatory postsynaptic current; RRP, readily releasable pool; SorCS1, Sortilin-related CNS expressed 1.

synaptic responses (Fig 5E and 5F) and RRP size (Fig 5G) induced by single application of 0.5 M sucrose were strongly decreased in *Sorcs1* KO neurons, in line with a reduction in active synapse number. Because eEPSC amplitude and RRP size were proportionally reduced in *Sorcs1* KO cells, the Pves (eEPSC charge/initial sucrose charge) was unchanged (Fig 5H). To further examine potential presynaptic release defects in *Sorcs1* KO neurons, we performed train stimulations at different frequencies to probe for short-term plasticity defects that cannot be detected by induction of single EPSC and single sucrose application. Repeated stimulation at 10 Hz produced a more pronounced rundown of normalized evoked responses (synaptic depression) in *Sorcs1* KO neurons compared with control cells (Fig 5I and 5J), suggesting a presynaptic defect during sustained periods of activity. Other stimulation frequencies and paired-pulse stimulations also showed increased synaptic depression in *SorCS1* KO neurons (Fig 5K). Calculating the RRP size by back-extrapolation of cumulative synchronous charge during 10 Hz train stimulation confirmed a reduction in RRP size in *Sorcs1* KO neurons (Fig 5L), indicating a presynaptic defect independent of the silent synapses phenotype that we described previously [20]. Together, these observations indicate that loss of SorCS1 impairs neurotransmitter release, reminiscent of Nrxn loss-of-function [5–7], indicating that SorCS1-mediated sorting of Nrxns in dendrites is required for normal presynaptic function.

## Discussion

The membrane trafficking mechanisms by which neurons control the polarized distribution and abundance of synaptic receptors are poorly understood. Here, we show that SorCS1-mediated sorting in dendritic endosomes controls a balance between axonal and dendritic surface levels of Nrxn and maintains Nrxn axonal surface polarization required for presynaptic differentiation and function.

### A SorCS1-Rab11FIP5/Rip11 interaction sorts axonal cargo from EEs to REs

The mechanism by which SorCS1 regulates trafficking of neuronal receptors has remained unclear. We find that SorCS1 controls a balance between axonal and dendritic surface polarization of its cargo Nrxn. In mature neurons, newly synthesized Nrxn1α traffics to the somato-dendritic surface, followed by endocytosis and transcytosis to the axon. Accordingly, blocking endocytosis, EE-, and RE-mediated transport all result in accumulation of Nrxn1α on the dendritic surface. These effects are mimicked by *Sorcs1* KO. We find that endosomal SorCS1 controls the transition of endocytosed Nrxn1α from EEs to REs and identify the Rab11-interacting protein Rab11FIP5/Rip11 as a novel SorCS1 cytoplasmic interactor. Interference with Rip11 function alters the axonal–dendritic surface balance of Nrxn1α in the same way that *Sorcs1* KO does. Rab11FIPs function as linkers between Rab11 and motor proteins to promote sorting of cargo from EEs to REs [37,47] and transcytosis in polarized epithelial cells [38,39], but their function in neuronal transcytosis has not been explored. Our data suggest that SorCS1/Rip11 form a protein complex that localizes to dendritic endosomes and sorts internalized Nrxn1α from early to recycling endosomes, thereby biasing the trafficking of Nrxn1α-

containing REs to the axon while preventing Nrxn1α missorting to the dendritic surface and lysosomes.

SorCS1-mediated sorting of Nrxn1α in dendrites can be circumvented by deleting the 4.1-binding motif in the cytoplasmic tail of Nrxn1α, which results in strong polarization of Nrxn1α to the axonal surface. Surface polarization of Nrxn1α Δ4.1 is unaffected by blocking endocytosis or removing *Sorcs1*. It remains to be determined whether Nrxn1α's 4.1-binding motif, which lacks canonical YXXØ and (D/E)*XXX*L(L/I) dendritic sorting motifs [48] but contains several tyrosine residues, is required for initial sorting to dendrites or for insertion in the dendritic membrane. Binding of the α-amino-3-hydroxy-5-methyl-4-isoxazole propionic-type receptor (AMPAR) subunit GluA1 to the 4.1N protein is required for GluA1 membrane insertion [49]. We find that the extracellular domain of Nrxn1α is required for somatodendritic sorting (S8D Fig and S8E Fig), suggesting that deletion of the 4.1-binding motif in Nrxn1α impairs dendritic insertion but not initial sorting to this compartment.

Our results show that loss of SorCS1 impairs transcytotic trafficking of Nrxn and L1/NgCAM, but not of Caspr2, which traffics via the selective endocytosis/retention pathway, supporting a role for SorCS1 in regulating transcytotic trafficking of axonal cargo proteins. It is unlikely that all membrane proteins that have been identified using surface proteome analysis to depend on SorCS1-3 for their surface trafficking [20,23,24] undergo transcytosis. The SorCS1 cargo amyloid precursor protein (APP) [50–52], however, is transcytosed in human neurons [53]. SorCS1 likely interacts with additional proteins, such as the retromer complex [50], to regulate the surface trafficking of a diverse set of cargoes.

## SorCS1-mediated surface polarization of Nrxn is required for presynaptic differentiation and function

The selective impairment in Nlgn1-induced presynaptic differentiation in *Sorcs1* KO neurons and rescue thereof by Nrxn1α Δ4.1, but not by WT Nrxn1α, indicates that this defect is due to a decrease in the abundance of axonal surface Nrxns. Nrxns play key roles in neurotransmitter release [5–7]. Alpha-Nrxn KO in neocortical cultured slices reduces mEPSC/miniature inhibitory postsynaptic current (mIPSC) frequency and evoked inhibitory postsynaptic current (eIPSC) amplitude [7]. β-Nrxn KO in cortical neurons decreases mEPSC frequency and eEPSC amplitude [5]. Our results show that loss of SorCS1 in autaptic cortical neurons decreases mEPSC frequency, eEPSC amplitude, and sucrose-evoked responses, in line with the reported decreased fraction of active synapses in *Sorcs1* KO neurons [20]. Whether this decrease in the fraction of active synapses is also happening in autaptic cultured neurons requires further experimental testing. Furthermore, we found an increase in synaptic depression of normalized eEPCSs during stimulus trains, similar to α-Nrxn KO neurons [7]. Together, these results suggest that mistrafficking of Nrxn is a major contributor to the defects in presynaptic function in *Sorcs1* KO neurons. These defects are likely attributed to a decrease in active synapse number and a presynaptic defect in neurotransmitter release, rather than a decrease in excitatory synapse density, which is unaffected in cultured *Sorcs1* KO neurons [20] and in Nrxn TKD hippocampal neurons [46].

The axonal–dendritic balance of Nrxn1α is developmentally regulated. Early in neuronal development, endogenous Nrxn1α is enriched on the axonal surface and traffics directly to the axon after trans-Golgi network (TGN) exit. As neurons mature, Nrxn1α follows an indirect trafficking pathway after TGN exit via SorCS1-mediated sorting in dendritic endosomes and transcytosis to the axonal surface. It will be interesting to determine whether expression of *Sorcs1* and *Rip11* is developmentally regulated in these neurons. The biological function of this developmental regulation and circuitous trafficking route is not clear. Transcytosis could

provide an efficient way of delivering receptors from a reservoir of readily synthesized proteins. Similar to other transcytosed axonal cargo, a dendritic pool of Nrxn may supply the demand for receptors along the axonal surface, either constitutively [54,55] or in response to ligand-receptor interactions and signalling [54,56].

Dendritically localized Nrxns might have a function in this compartment. Postsynaptically expressed Nrxn1 decreases Nlgn1's synaptogenic effect in hippocampal neurons, likely by *cis*-inhibition of Nlgn1 [32]. In retinal ganglion cells, a shift in Nrxn localization away from dendrites has been proposed to allow dendritic innervation [30]. In cortical neurons, however, we observe the opposite: Nrxn surface expression in dendrites increases with maturation. Dendritic Nrxns might modulate the function of postsynaptic Nrxn ligands, such as Nlgns, which control postsynaptic neurotransmitter receptor function [57], or leucine-rich repeat transmembrane neuronal protein 1 (LRRTM1), which shapes presynaptic properties [58]. At the *Caenorhabditis elegans* neuromuscular junction, the ectodomain of postsynaptic Nrxn is proteolytically cleaved and binds to the presynaptic α2δ calcium channel subunit to inhibit presynaptic release [59].

In conclusion, impaired trafficking of critical cargo proteins in the absence of SorCS1 and resulting defects in synaptic function might contribute to the pathophysiology of the neurodevelopmental [60–63] and neurodegenerative disorders [52,64] with which *SORCS1* has been associated. These observations underscore the notion that intracellular sorting is a key contributor to the proper maintenance of synaptic protein composition and function.

## Materials and methods

### Ethics statement

All animal procedures were approved by the Institutional Animal Care and Research Advisory Committee of the KU Leuven (ECD P015/2013 and ECD P214/2017) and were performed in accordance with the Animal Welfare Committee guidelines of the KU Leuven, Belgium. The health and welfare of the animals were supervised by a designated veterinarian. The KU Leuven animal facilities comply with all appropriate standards (cages, space per animal, temperature, light, humidity, food, water), and all cages are enriched with materials that allow the animals to exert their natural behaviour. Both males and females were used for all experiments. To the best of our knowledge, we are not aware of an influence of sex on the parameters analyzed in this study.

### Animals

Wistar rats were obtained from Janvier Labs (Le Genest-Saint-Isle, France). CD-1 and C57BL/6J mice were obtained from the KU Leuven Mouse Facility. C57BL/6J *Sorcs1*<sup>flox/flox</sup> mice were previously described [50] and kindly provided by Alan D. Attie (University of Wisconsin-Madison, USA). C57BL/6J *Nrxn1α*<sup>HA</sup> and C57BL/6J *Sorcs1*<sup>HA</sup> KI mice were generated by CRISPR/Cas9-mediated homology directed repair (HDR) targeting the mouse *Nrxn1* locus at exon 2, and *Sorcs1* locus at exon 1. *Sorcs1*<sup>HA</sup> was commercially purchased from Cyagen (Santa Clara, CA), whereas *Nrxn1α*<sup>HA</sup> was generated in-house at the Mutamouse core facility (KU Leuven/VIB). To generate *Nrxn1α*<sup>HA</sup> KI, CRISPR/Cas9 components were microinjected in zygotes as ribonucleoproteins (RNPs) composed of: crRNA (guide RNA sequence: 5′-TCCACTGGCCCTCGGCGCCC-3′; IDT, Coralville, IA), tracrRNA (IDT, Coralville, IA, Cat #1072534), ssODN (oligo DNA donor sequence: 5′-TGCTGTCTCCTCTGCCTGTCGCTG CTGCTGCTGGGCTGCTGGGCAGAGCTGGGCAGCGGGTACCCATACGACGTCCCAG ATTACGCTCTGGAGTTTCCGGGCGCCGAGGGCCAGTGGACGCGCTTCCCCAAGTG GAACGCGTGCTGC-3′; IDT, Coralville, IA) and Cas9 (IDT, Coralville, IA, Cat #1074182).

To generate *Sorcs1^HA* KI, the donor DNA (5′-GTCGGAGCCGCCGGGACATGCTAAAGG ATGGAGGGCAGCAGGGGCTTGGGACTGGCGCATACCCATACGATGTTCCAGATT ACGCTCGGGACCCGGACAAAGCCACCCGCTTCCGGATGGAGGAGCTGAGACTG ACCAGCACCACA-3′) and guide RNA (5′-TGGGACTGGCGCACGGGACCCGG-3′) were used. Insertion of the HA epitope tag was validated by sequencing, restriction digest (for *Nrxn1α^HA*), and western blot. To mitigate possible CRISPR/Cas9-mediated off-targets, *Nrxn1α^HA* and *Sorcs1^HA* mice were backcrossed to WT C57BL/6J. *Nrxn1α^HA* and C57BL/6J *Sorcs1^HA* mice are fertile and viable.

## Cell lines

HEK293T-17 human embryonic kidney cells (available source material information: foetus) were obtained from American Type Culture Collection (ATCC, Manassas, VA, Cat #CRL-11268). HEK293T-17 cells were grown in DMEM (Thermo Fisher Scientific, Waltham, MA, Cat #11965092) supplemented with 10% (vol/vol) FBS (Thermo Fisher Scientific, Waltham, MA Cat, #10270106) and 1% (vol/vol) penicillin/streptomycin (Thermo Fisher Scientific, Waltham, MA, Cat #15140122).

## Neuronal cultures

**Primary neuronal cultures for imaging.** Neurons were cultured from E18 to E19 Wistar rat embryos (cortical), from E18 to E19 WT C57BL/6J mice embryos (cortical and hippocampal), from E18 to E19 *Sorcs1^{flox/flox}* mice (cortical), from E18 to E19 *Sorcs1^HA* mice (cortical), and E18 to E19 *Nrxn1α^HA* mice (cortical), as previously described by Savas and colleagues [20]. Briefly, dissected hippocampi and cortices were incubated with trypsin (0.25% [vol/vol], 15 min, 37˚C; Thermo Fisher Scientific, Waltham, MA, Cat #15090046) in HBSS (Thermo Fisher Scientific, Waltham, MA, Cat #14175095) supplemented with 10 mM HEPES (Thermo Fisher Scientific, Waltham, MA, Cat #15630080). After trypsin incubation, hippocampi and cortices were washed with MEM (Thermo Fisher Scientific, Waltham, MA, Cat #11095080) supplemented with 10% (v/v) horse serum (Thermo Fisher Scientific, Waltham, MA, Cat #26050088) and 0.6% (wt/vol) glucose (MEM-horse serum medium) 3 times. The cells were mechanically dissociated by repeatedly pipetting the tissue up and down in a flame-polished Pasteur pipette and then plated on poly-D-lysine (Millipore, Burlington, MA, Cat #A-003-E) and laminin (Sigma-Aldrich, St. Louis, MO, Cat #L2020) coated glass coverslips (Glaswarenfabrik Karl Hecht, Sondheim vor der Rhön, Germany, Cat #41001118) in 60 mm culture dishes (final density: $4 \times 10^5 - 7 \times 10^5$ cells per dish), containing MEM-horse serum medium. Once neurons attached to the substrate, after 2 to 4 h, the coverslips were flipped over an astroglial feeder layer in 60-mm culture dishes containing neuronal culture medium: Neurobasal medium (Thermo Fisher Scientific, Waltham, MA, Cat #21103049) supplemented with B27 (1:50 dilution; Thermo Fisher Scientific, Waltham, MA, Cat #17504044), 12 mM glucose, glutamax (1:400 dilution; Thermo Fisher Scientific, Waltham, MA, Cat #35050061), penicillin/ streptomycin (1:500 dilution; Thermo Fisher Scientific, Waltham, MA, Cat #15140122), 25 µM β-mercaptoethanol, and 20 µg/mL insulin (Sigma-Aldrich, St. Louis, MO, Cat #I9278). Neurons grew face down over the feeder layer but were kept separate from the glia by wax dots on the neuronal side of the coverslips. To prevent overgrowth of glia, neuron cultures were treated with 10 µM 5-Fluoro-2′-deoxyuridine (Sigma-Aldrich, St. Louis, MO, Cat #F0503) after 3 d. Cultures were maintained in a humidified incubator of 5% (vol/vol) $CO_2$/95% (vol/vol) air at 37˚C, feeding the cells once per week by replacing one-third of the medium in the dish. Cortical neurons from WT and *Sorcs1^{flox/flox}* mice were electroporated with DNA just before plating using an AMAXA Nucleofector kit (Lonza, Basel, Switzerland, Cat #VPG-1001). Cortical

neurons from *Sorcs1*$^{flox/flox}$ mice were transfected at DIV7 using Effectene (Qiagen, Hilden, Germany, Cat #301425). Rat cortical neurons and neurons from WT (cortical and hippocampal) and *Sorcs1*$^{flox/flox}$ (cortical) mice were transfected at DIV6 to DIV8 using calcium phosphate. Mouse cortical neurons from *Nrxn1α*$^{HA}$ and *Sorcs1*$^{HA}$ were mixed with WT cortical neurons which allowed for reliable immunodetection of endogenous HA-Nrxn1α and HA-SorCS1.

Co-culture assays were performed as previously described by de Wit and colleagues [65]. Briefly, HEK293T cells grown in DMEM (Thermo Fisher Scientific, Waltham, MA, Cat #11965092) supplemented with 10% (vol/vol) FBS (Thermo Fisher Scientific, Waltham, MA, Cat #10270106) and 1% (vol/vol) penicillin/streptomycin (Thermo Fisher Scientific, Waltham, MA, Cat #15140122) were transfected with FLAG-tagged Nlgn1 or EGFP-tagged NGL-3 using Fugene6 (Promega, Madison, WI, Cat #E2691). Transfected HEK293T cells were mechanically dissociated 24 h after transfection and co-cultured with DIV10 cortical neurons prepared from *Sorcs1*$^{flox/flox}$ mice for 16 hr.

**Primary autaptic cortical cultures for electrophysiology.** For electrophysiology, we used isolated cortical neurons on astrocyte microislands. In short, cortices were dissected from E18 *Sorcs1*$^{flox/flox}$ mice and collected in HBSS (Thermo Fisher Scientific, Waltham, MA, Cat #14175095) supplemented with 10 mM HEPES (Thermo Fisher Scientific, Waltham, MA, Cat #15630080). Cortical pieces were incubated for 15 min in HBSS supplemented with trypsin 0.25% (vol/vol) (Thermo Fisher Scientific, Waltham, MA, Cat #15090046) and 10 mM HEPES for 15 min at 37˚C. After trypsin blocking and washing with MEM (Thermo Fisher Scientific, Waltham, MA, Cat #11095080) supplemented with 10% (v/v) horse serum (Thermo Fisher Scientific, Waltham, MA, Cat #26050088) and 0.6% (wt/vol) glucose; neurons were dissociated, counted, and plated in Neurobasal medium (Thermo Fisher Scientific, Waltham, MA, Cat #21103049) supplemented with B27 (1:50 dilution; Thermo Fisher Scientific, Waltham, MA, Cat #17504044), 12 mM glucose, glutamax (1:400 dilution; Thermo Fisher Scientific, Waltham, MA, Cat #35050061), penicillin/streptomycin (1:500 dilution; Thermo Fisher Scientific, Waltham, MA, Cat #15140122), and 25 μM β-mercaptoethanol. Dissociated neurons were electroporated (*Sorcs1*$^{flox/flox}$: EGFP or Cre-EGFP) just before plating using AMAXA Nucleofector kit (Lonza, Basel, Switzerland, Cat #VPG-1001) and plated at 2,500/cm$^2$ on micro islands of mouse (CD-1) glia. Glial islands were obtained by first coating glass coverslips with 0.15% (wt/vol) agarose. After drying and UV sterilization (30 min), customized stamps were used to create dots (islands, diameter 200–250 μm) using a substrate mixture containing 0.25 mg/mL rat tail collagen (Corning, Corning, NY, Cat #354236) and 0.4 mg/mL poly-D-lysine (Sigma-Aldrich, St. Louis, MO, Cat #P7405) dissolved in 17 mM acetic acid. After drying/UV treatment of the islands, approximately 25,000 astrocytes were plated per 30 mm glass coverslip in 6-well plates and allowed to form micro-dot islands in DMEM (Thermo Fisher Scientific, Waltham, MA, Cat #11965092) supplemented with 10% (vol/vol) FBS (Thermo Fisher Scientific, Waltham, MA, Cat #10270106) for 3 to 5 d.

## Plasmids

Constructs were all generated using the Gibson Assembly Cloning Kit (New England Biolabs, Ipswich, MA, Cat #E5510S) by inserting the different DNA fragments (PCR-generated, gBlocks, or ultramers) in the final vectors digested with the respective restriction enzymes. pCAG-HA-Nrxn1α(-SS4) was kindly provided by Peter Scheiffele (University of Basel, Switzerland). For pFUGW-mCherry, pFUGW-Cre_T2A_mCherry, pFUGW-Cre_T2A_mCherry_T2A_Nrxn1α(-SS4) WT, pFUGW-Cre_T2A_mCherry_T2A_Nrxn1α(-SS4) Δ4.1, pFUGW-Cre-EGFP, pFUGW-Cre-EGFP_T2A_SorCS1$^{WT}$, pFUGW-Cre-EGFP_T2A_Sor

CS1$^{Y1132A}$, pFUGW-EGFP_T2A_HANrxn1α(-SS4), and pFUGW-EGFP_T2A_HA-Nrxn1α (-SS4)ΔN-terminal (remaining N terminal consists of HA epitope tag and 5 amino acids before the transmembrane region) PCR fragments were inserted into pFUGW vector digested with AgeI and EcoRI. For pcDNA3.1-HA-SorCS1cβ, a PCR fragment was inserted into pcDNA3.1 (+) (Thermo Fisher Scientific, Waltham, MA, Cat #V79020) digested with EcoRI. For FCK (0.4)GW-Cre-EGFP a PCR fragment was inserted into FCK(0.4)GW vector digested with AgeI and EcoRI. For pIRESneo3-Str-KDEL_SBP-EGFP-Nrxn1α, a PCR fragment was inserted into pIRESneo3-Str-KDEL_SBP-EGFP-GPI (kindly provided by Franck Perez, Institute Curie, France; Addgene, Watertown, MA, plasmid #65294; RRID: Addgene_65294) digested with FseI and PacI. pcDNA4-SorCS1cβ-myc Y1132A, pcDNA3.1-HA-SorCS1cβ Y1132A and EGFP-tagged Rab7 T22N were generated by in vitro mutagenesis using the QuikChange II Site-Directed Mutagenesis Kit (Agilent, Santa Clara, CA, Cat #200522) from pcDNA4-SorCS1cβ-myc (kindly provided by Alan D. Attie, University of Wisconsin-Madison, USA), pcDNA3.1-HA-SorCS1cβ and EGFP-tagged Rab7 WT (kindly provided by Casper Hoogenraad, Utrecht University, The Netherlands), respectively. The following constructs were kindly provided: pIRESneo3-Str-KDEL_TfR-SBP-EGFP and EGFP-tagged TfR (Juan S. Bonifacino, National Institute of Health, USA); pEGFP-N3-Rip11 and pEGFP-C1-Rip11(490–653) (Rytis Prekeris, University of Colorado, USA); EGFP-tagged Rab11 WT, EGFP-tagged Rab11 S25N and EGFP-tagged Rab4 WT (Casper Hoogenraad, Utrecht University, The Netherlands); EGFP-tagged Rab5 WT and EGFP-tagged Rab5 S34N (Ragna Sannerud, KU Leuven, Belgium); EGFP-tagged Dynamin1 WT, EGFP-tagged Dynamin1 WT K44A [66] and pcDNA3-HA-Caspr2 (Catherine Faivre-Sarrailh, Aix-Marseille University, France); L315 control and L315 Nrxns TKD (Thomas C. Südhof, Stanford University, USA); FLAG-tagged Nlgn1 (Davide Comoletti, Rutgers University, USA); pEGFP-N1-NGL-3 (Eunjoon Kim, Korea Advanced Institute of Science and Technology, South Korea); and myc-tagged L1 [67] (Dan P. Felsenfeld, CHDI Foundation, USA). All DNA constructs used in this study were verified by sequencing. See S1 Table for a list of plasmids used in this study.

## Neuron transfection

Rat cortical neurons (live-cell imaging experiments) and neurons from WT (cortical and hippocampal) and *Sorcs1*$^{flox/flox}$ (cortical) mice were transfected at DIV6 to DIV8 using calcium phosphate, with the exception of the DIV3 live-cell imaging experiment, in which rat cortical neurons were transfected at DIV2. A total of 2 μg of DNA or 1 μg of DNA from each DNA construct (double co-transfections) were used per coverslip, with the exception of co-transfections of EGFP-tagged Rab proteins or Rip11 (WT and dominant negatives) with extracellular HA-tagged Nrxn1α, in which 0.75 μg and 1.25 μg of DNA was used for EGFP-tagged and HA-Nrxn1α DNA constructs, respectively. Briefly, DNA plasmids were diluted in Tris-EDTA buffer (10 mM Tris-HCl and 2.5 mM EDTA [pH 7.3]), followed by dropwise addition of CaCl$_2$ solution (2.5 M CaCl$_2$ in 10 mM HEPES [pH 7.2]) to the plasmid DNA-containing solution to give a final concentration of 250 mM CaCl$_2$. This solution was subsequently added to an equal volume of HEPES-buffered solution (274 mM NaCl, 10 mM KCl, 1.4 mM Na$_2$HPO$_4$, 42 mM HEPES [pH 7.2]) and vortexed gently for 3 s. This mixture, containing precipitated DNA, was then added dropwise to the coverslips in a 12-well plate, containing 250 μL of conditioned neuronal culture medium with kynurenic acid (2 mM), followed by 2 h incubation in a 37˚C, 5% (vol/vol) CO$_2$/95% (vol/vol) air incubator. After 2 h, the transfection solution was removed, after which 1 mL of conditioned neuronal culture medium with kynurenic acid (2 mM) slightly acidified with HCl (approximately 5 mM final concentration) was added to each coverslip, and the plate was returned to a 37˚C, 5% (vol/vol) CO$_2$/95% (vol/vol) air incubator

for 20 min. Finally, coverslips were then transferred to the original dish containing the conditioned culture medium in the 37˚C, 5% (vol/vol) $CO_2$/95% (vol/vol) air incubator to allow expression of the transfected constructs. Protein expression was typically for 24 h (live-cell imaging) or 48 h (immunocytochemistry).

## Immunocytochemistry

Cells were fixed for 10 to 15 min in 4% (wt/vol) sucrose and 4% (wt/vol) paraformaldehyde in PBS (PBS: 137 mM NaCl, 2.7 mM KCl, 1.8 mM $KH_2PO_4$ and 10 mM $Na_2HPO_4$ [pH 7.4]) at room temperature and permeabilized with PBS + 0.25% (vol/vol) Triton X-100 for 5 min at 4˚C. Neurons were then incubated in 10% (wt/vol) bovine serum albumin (BSA) in PBS for 1 h at room temperature to block nonspecific staining and incubated in appropriate primary antibodies diluted in 3% (wt/vol) BSA in PBS (overnight, 4˚C). After washing 3 times in PBS, cells were incubated with the secondary antibodies diluted in 3% (wt/vol) BSA in PBS (1 h, room temperature). The coverslips were mounted using Prolong Gold Antifade mounting medium (Thermo Fisher Scientific, Waltham, MA, Cat #P36930).

In all immunocytochemistry experiments performed to determine the subcellular distribution of proteins of interest (Nrxns, SorCS1, Caspr2, L1) the axonal compartment was labeled using antibodies against Ankyrin-G, a marker of the AIS (NeuroMab, Davis, CA, Cat #73–146 or Santa Cruz Biotechnology, Dallas, TX, Cat #sc-31778), and the somatodendritic compartment was labeled using an anti-MAP2 antibody (Abcam, Cambridge, England, Cat #ab5392). See S2 Table for a list of antibodies used in this study.

**Surface immunostaining of HA-tagged proteins.** For surface immunostaining of exogenous HA-Nrxn1α and HA-Caspr2, live mouse cortical and hippocampal neurons were incubated with rabbit anti-HA (1:1,000 dilution; Sigma-Aldrich, St. Louis, MO, Cat #H6908) diluted in conditioned neuronal culture medium for 15 min at room temperature. For surface immunostaining of endogenous HA-Nrxn1α, live mouse cortical $Nrxn1\alpha^{HA}$ KI neurons were incubated with rabbit anti-HA (1:100 dilution; Cell Signaling Technology, Danvers, MA, Cat #3724) diluted in conditioned neuronal culture medium for 20 min at room temperature. Neurons were then fixed in 4% (wt/vol) sucrose and 4% (wt/vol) paraformaldehyde in PBS for 10 min at room temperature, followed by several washes in PBS and blocking in 10% (wt/vol) BSA in PBS for 1 h at room temperature. Neurons were then incubated with anti-rabbit secondary antibody diluted in 3% (wt/vol) BSA in PBS (1 h, room temperature). Following permeabilization, neurons were processed for immunocytochemistry as described above.

**Surface immunostaining of extracellular myc-tagged L1.** To allow surface immunostaining of myc-L1 (live-labeling proved to be impossible), neurons were fixed first, followed by several washes and blocking, and then incubated with mouse anti-myc (1:1,000 dilution; Santa Cruz Biotechnology, Dallas, TX, Cat #sc-40) diluted in 3% (wt/vol) BSA in PBS overnight at 4˚C. Subsequently, neurons were incubated with the respective secondary antibody for 1 h at room temperature and processed for immunocytochemistry as described above.

**Antibody pulse-chase experiments.** Cultured living neurons were incubated at room temperature for 10 min in the presence of a high concentration (1:250) of mouse anti-HA antibody (Covance, Princeton, NJ, Cat #MMS-101P) against extracellular HA-tagged Nrxn1α and SorCS1, diluted in conditioned medium. Neurons were then washed with prewarmed PBS at 37˚C to remove the unbound antibody and were further incubated in antibody free conditioned medium in a 37˚C, 5% (vol/vol) $CO_2$/95% (vol/vol) air incubator (for different periods) to allow the internalization of antibody-bound receptors. After this incubation, neurons were fixed in 4% (wt/vol) sucrose and 4% (wt/vol) paraformaldehyde in PBS for 10 min at room temperature. Next, neurons were either exposed to a supersaturating concentration (1:300) of

the first of 2 secondary antibodies, to label the primary antibody-bound surface pool of protein, and/or incubated overnight with anti-mouse Fab fragments (0.25 mg/mL; Santa Cruz Biotechnology, Dallas, TX, Cat #715-007-003) in 5% (wt/vol) BSA in PBS to block all primary antibody-bound receptors that were not internalized and/or not labeled by the first secondary antibody. After permeabilization, cells were processed for immunocytochemistry as described above, and the pool of internalized receptors was labeled by incubation with the second secondary antibody (1:1,000) for 1 h at room temperature. This strategy allows differential labeling of cell surface and internalized pools of protein.

**Dynasore treatment.**   Mouse cortical neurons expressing extracellular HA-tagged Nrxn1α for 30 h were either treated with 20 μM Dynasore (Sigma-Aldrich, St. Louis, MO, Cat #D7693) or DMSO (vehicle) and added to the coverslips in a 12-well plate containing conditioned neuronal culture medium for 18 h. Afterwards, surface HA-Nrxn1α was labeled in live neurons as described above.

## Image analysis and quantification

All images obtained from immunocytochemistry experiments in fixed cells were captured on a Leica SP8 laser-scanning confocal microscope (Leica Micro-systems, Wetzlar, Germany). The same confocal acquisition settings were applied to all images taken from a single experiment. Parameters were adjusted so that the pixel intensities were below saturation. Fiji analysis software was used for quantitative imaging analysis. Z-stacked images were converted to maximal intensity projections and thresholded using constant settings per experiment.

**Quantification of axonal and dendritic immunofluorescence intensity and ratio of axonal–dendritic immunofluorescence intensity.**   Fluorescence intensity was measured as the sum of integrated intensity in representative portions of axons and dendrites using Ankyrin-G and MAP2 as guides, respectively. Axonal and dendritic intensities were divided by neuritic length and total intensity (axonal + dendritic) ('Relative Axonal Intensity' and 'Relative Dendritic Intensity') to adjust measurements across cells with varying expression levels (exogenous expression). The A–D ratio was then calculated by dividing the values of axonal and dendritic intensities obtained for every cell ('Axonal/Dendritic Intensity'). A uniformly distributed protein yields an A–D ratio of around 1. A preferentially dendritically localized protein yields an A–D ratio < 1, whereas a preferentially axonally localized protein yields an A–D ratio > 1. To quantify the fluorescence intensity of endogenous Nrxns, axonal and dendritic intensities were not normalized to the total intensity ('Axonal Intensity' and 'Dendritic Intensity').

**Polarity index.**   In some cases, polarization of cargo was determined by using the classical polarity index originally described by Sampo and colleagues and Wisco and colleagues [68,69]. This index does not provide a direct measurement of the axonal and dendritic fluorescent intensities, but it allows a quick estimation of the compartmentalized polarization of proteins of interest. One-pixel-wide lines were traced along 3 dendrites and representative portions of the axon, using MAP2 and Ankyrin-G as guides, in nonthresholded images. The mean intensities (dendritic was averaged from 3 dendrites) were used to calculate the dendrite:axon (D:A) 'Polarity Index'. D:A = 1, uniform staining; D:A < 1, preferential axonal staining; D:A > 1, preferential dendritic staining.

**Long-term live-cell imaging.**   Fluorescence intensity was measured as the sum of integrated intensity in representative portions of axons and dendrites (using Ankyrin-G and MAP2 as guides) and cell bodies. Axonal and dendritic intensities were divided by neuritic length and somatic intensities by somatic area ('Nrxn Intensity' or 'TfR Intensity').

**Antibody pulse-chase experiments.** Fluorescence intensity was measured as the sum of integrated intensity in representative portions of axons and dendrites using Ankyrin-G and MAP2 as guides, respectively. Intensities of internalized SorCS1 and Nrxn were divided by neuritic length and by the total intensity (surface + internal) ('Relative Intensity of Internalized Nrxn' or 'SorCS') or by the total internalized intensity (axonal + dendritic) ('Relative Intensity of Internalized Nrxn'). Intensities of surface SorCS1 were also divided by neuritic length and by the total intensity (surface + internal) ('Relative Intensity of Surface SorCS1').

**Manders coefficient.** Manders coefficient was measured by using the Fiji plugin JACoP [70]. Manders coefficient measures the proportion of the signal from channel 'a' that coincides with the signal in channel 'b' over the total intensity of 'a'–M1 coefficient [71].

**Colocalization of internalized Nrxn with endosomal markers in Sorcs1 KO cells.** The colocalization of internalized Nrxn with endosomal markers was evaluated by measuring the density of double-positive puncta for internalized Nrxn and endosomes and by measuring the intensity of Nrxn present in these puncta. Fluorescence intensity was measured as the sum of mean intensity of internalized Nrxn puncta (defined as 0.02 $\mu m^2$-infinite) in representative portions of axons and dendrites (using Ankyrin-G and MAP2 as guides), colocalizing with endosomal markers. Intensities of internalized Nrxn were divided by neuritic length ('Nrxn Intensity in Double Positive Puncta'). Density of internalized Nrxn puncta (defined as 0.02 $\mu m^2$-infinite), colocalizing with endosomal markers, was also divided by neuritic length ('Double Positive Puncta Density').

## Live-cell imaging

For live-cell imaging of fluorescently tagged Nrxn1α- and TfR-positive vesicles, neurons were transfected with a bicistronic expression plasmid encoding Streptavidin-KDEL and SBP-EGFP-Nrxn1α or Streptavidin-KDEL and TfR-SBP-EGFP, respectively, using the RUSH system [34]. After transfection, neurons were maintained in Neurobasal medium without B27 supplement, because the presence of D-biotin in B27 interferes with the RUSH system, or in Neurobasal medium supplemented with N-2 supplement (Thermo Fisher Scientific, Waltham, MA, Cat #P36930), which lacks D-biotin. Live-cell imaging was performed at room temperature (approximately 20˚C) in imaging medium (119 mM NaCl, 5 mM KCl, 2 mM CaCl2, 2 mM MgCl2, 30 mM glucose, and 10 mM Hepes [pH 7.4]). For long-term live-cell imaging experiments DIV8 to DIV10 or DIV3 rat cortical neurons, after 24 to 31 h or 25 to 29 h of protein expression, respectively, were imaged on a Nikon Eclipse Ti A1R confocal microscope (Minato, Japan). Images were collected with a 40× oil objective (1.3 NA; Nikon, Minato, Japan). Focal drift during the experiment was avoided by using the perfect focus system feature of the Nikon system. Laser intensities were kept as low as possible. Time lapses lasted for 2.5 h and frames were acquired as z-stacks every 5 min. Synchronous release of Nrxn1α or TfR was induced by application of 40 μM D-biotin (Sigma-Aldrich, St. Louis, MO, Cat #B4501) 10 min after the beginning of the imaging session. Protein synthesis was inhibited by including 20 μg/mL cycloheximide (Sigma-Aldrich, St. Louis, MO, Cat #C4859) in the imaging medium. For short-term live-cell imaging experiments, DIV8 to DIV10 rat cortical neurons expressing Streptavidin-KDEL and SBP-EGFP-Nrxn1α, 21 to 30 h after protein expression, were imaged on a Nikon spinning-disk confocal microscope equipped with a 60× oil objective (1.4 NA; Nikon). Time-lapses were performed by sequential capture of 200 ms images of SBP-EGFP-Nrxn1α every second for 60 to 120 s. Focal drift during the experiment was corrected automatically using the autofocus feature of the Nikon system. Neurons were imaged either for 20 to 34 min or 1 to 2 h after exposure to 40 μM D-biotin. In both experiments, a single image of live-stained pan-Neurofascin was taken at the beginning of every imaging session.

**Live-labeling of the AIS.** Before every live-cell imaging experiment, primary rat cortical cultured neurons were live-labeled with an anti-pan-Neurofascin antibody (NeuroMab, Davis, CA, Cat #75–172) to distinguish the axon from the dendrites. Live-cell imaging experiments were performed with rat cortical neurons because live-labeling of the AIS with the anti-pan-Neurofascin antibody did not work in mouse cultured neurons in our hands. Briefly, cover-slips with neurons were quickly rinsed in prewarmed neuronal culture medium. Neurons were incubated with anti-pan-Neurofascin antibody (1/500) diluted in conditioned neuronal culture medium for 10 min in a 37°C, 5% (vol/vol) $CO_2$/95% (vol/vol) air incubator. Coverslips were then quickly washed twice with prewarmed neuronal culture medium. Finally, neurons were incubated with an Alexa-555 anti-mouse (1/400; Thermo Fisher Scientific, Waltham, MA, Cat #A31570) secondary antibody diluted in conditioned neuronal culture medium and incubated for 10 min in a 37°C, 5% (vol/vol) $CO_2$/95% (vol/vol) air incubator. Neurons were quickly rinsed twice with prewarmed neuronal culture medium and used for live-cell imaging experiments.

**Analysis of Nrxn1α- and TfR-positive vesicle transport.** Dendrites and axon were imaged from the same neuron using spinning-disk confocal microscopy. Time lapses were performed by sequential capture of 200 ms images every second for 60 to 120 s. Acquisitions were analyzed using an Igor-based software developed by Pieter Vanden Berghe and Valérie Van Steenbergen (Lab. for Enteric NeuroScience and Cell Imaging Core, KU Leuven). Kymographs were created using a segmented line along the axon/dendrite (1 dendrite was analyzed per cell) from the soma towards the neurite tip, so that anterograde movement occurred from left to right. On the kymographs, single vesicle movement episodes were distinguished as tilted straight lines. Single vesicle pauses were distinguished as straight lines. The number of moving anterograde/retrograde particles in the dendrite and axon was determined manually by drawing lines on top of the trajectories of single particles obtained from the kymographs. Vesicles were defined as motile when showing net displacement of $\geq 2$ μm.

## Biochemistry

**Detection of HA-tagged endogenous SorCS1 and Nrxn1α.** Adult brains ($Sorcs1^{HA}$, P60 total brain) or ($Nrxn1\alpha^{HA}$, 3-month-old cortices) from WT(+/+), heterozygous (+/Tg), and homozygous (Tg/Tg) KI mice were dissected, snap frozen, and stored at −80°C. The tissue was homogenized in homogenization buffer (20 mM HEPES [pH 7.4], 320 mM sucrose, 5 mM EDTA) supplemented with protease inhibitors (Roche, Basel, Switzerland, Cat #11697498001) using a glass Dounce homogenizer. Homogenates were spun at 3,000$g$ for 15 min at 4°C. Supernatants were collected, and protein quantification was performed with Bio-Rad protein Assay (Bio-Rad, Hercules, CA, Cat #500–0006). Samples were analyzed by western blot using a mouse anti-HA antibody (Covance, Princeton, NJ, Cat #MMS-101P; $Sorcs1^{HA}$ KI) or a rabbit anti-HA antibody (Cell Signaling Technology, Danvers, MA, Cat #3724; $Nrxn1\alpha^{HA}$ KI). A total protein stain (Ponceau) was used to assess equal protein loading and transfer to nitrocellulose membrane.

**Immunoprecipitation of HA-tagged endogenous SorCS1.** Cortices from adult $Sorcs1^{HA}$ KI mouse brains were dissected and homogenized in homogenization buffer (50 mM HEPES [pH 7.4], 100 mM NaCl, 2 mM $CaCl_2$, 2.5 mM $MgCl_2$) supplemented with protease inhibitors (Roche, Basel, Switzerland, Cat #11697498001) using a Dounce homogenizer. Cortical homogenates were extracted with 1% CHAPSO (VWR, Radnor, PA, Cat #A1100.0005) in homogenization buffer while rotating end-over-end for 2 h at 4°C and centrifuged at 100,000$g$ for 1 h at 4°C to pellet insoluble material. Supernatants were precleared by adding 100 μL of Protein-G agarose beads (Thermo Fisher Scientific, Waltham, MA, Cat #22852BR) and rotating for 1 h at

4˚C, then incubated with 5 μg of mouse IgG or 50 μL of anti-HA magnetic beads (Thermo Fisher Scientific, Waltham, MA, Cat #88836) and rotated end-over-end overnight at 4˚C. A total of 60 μL of Protein-G agarose beads were added to the IgG control sample and rotated for 1 h at 4˚C, followed by 3 washes in cold extraction buffer and once in PBS. Anti-HA magnetic beads were washed according to the manufacturer's protocol using a magnetic stand. Protein-G agarose beads and anti-HA magnetic beads were heated for 15 min at 40˚C in 50 μL 2X sample buffer and analyzed by western blotting.

## LV production

Second generation VSV.G pseudotyped LVs were produced as described by Dittgen and colleagues and Kutner and colleagues [72,73]. HEK293T cells were transfected with control (mCherry) or Cre-T2A-mCherry-containing pFUGW vector plasmids and helper plasmids PAX2 and VSVG using Fugene6 (Promega, Madison, WI, Cat #E2691). Supernatant was collected 65 h after transfection and filtered through a 0.45 μm filter (Thermo Fisher Scientific, Waltham, MA, Cat #723–2545), aliquoted, and stored at −80˚C.

## Electrophysiology

Neurons were recorded on DIV14 to DIV16. The patch pipette solution contained (in mM): 136 KCl, 18 HEPES, 4 Na-ATP, 4.6 MgCl$_2$, 4 K$_2$-ATP, 15 Creatine Phosphate, 1 EGTA, and 50 U/ml Phospocreatine Kinase (300 mOsm [pH 7.30]). The external medium used contained the following components (in mM): 140 NaCl, 2.4 KCl, 4 CaCl$_2$, 4 MgCl$_2$, 10 HEPES, 10 Glucose (300 mOsm [pH 7.30]). Cells were whole-cell voltage clamped at −70 mV with a double EPC-10 amplifier (HEKA Elektronik, Lambrecht/Pfalz, Germany) under control of Patchmaster v2x32 software (HEKA Elektronik, Lambrecht/Pfalz, Germany). Currents were low-pass filtered at 3 kHz and stored at 20 kHz. Patch pipettes were pulled from borosilicate glass using a multistep puller (P-1000; Sutter Instruments, Novato, CA). Pipette resistance ranged from 3 to 5 MΩ. The series resistance was compensated to approximately 75%. Only cells with series resistances below 15 MΩ were included for analysis. All recordings were made at room temperature. Spontaneous glutamatergic release (sEPSC) was recorded at −70 mV. Evoked release was induced using brief depolarization of the cell soma (from 70 to 0 mV for 1 ms) to initiate action potential-dependent glutamatergic release (eEPSCs). A fast local multibarrel perfusion system (Warner SF-77B, Warner Instruments, Hamden, CT) was used to determine the RRP size using external recording solution containing 500 mM sucrose. A custom analysis procedure in Igor Pro (Wavemetrics Inc., Lake Oswego, OR) was used for offline analysis of evoked and sucrose responses. Spontaneous events were detected using Mini Analysis program (Synaptosoft, Fort Lee, NJ).

## Statistical analysis

Results are shown as average or as average ± SEM, with *n* referring to the number of analyzed neurons for each group. For most experiments at least 3 independent cultures were included for analysis. Data sets were tested either using Mann-Whitney *U* test or Kruskal-Wallis test by Dunn's multiple comparisons test. Statistical testing was performed using GraphPad Prism (GraphPad Software, San Diego, CA). In all instances, statistical significance was defined as follows: NS, not significant ($p > 0.05$); *$p < 0.05$; ** $p < 0.01$; *** $p < 0.001$.

## Supporting information

**S1 Movie. DIV8 WT rat cortical neuron co-expressing the ER hook (Streptavidin-KDEL) and SBP-EGFP-Nrxn1α (grayscale, inverted for clarity) and live-stained for the AIS marker Neurofascin to label the axon.** Biotin was added 10 min after the beginning of the imaging session. Cell was recorded every 5 min for 2.5 h. The axon is indicated. Frame rate: 2 fps. AIS, axon initial segment; DIV, days in vitro; ER, endoplasmic reticulum; EGFP, enhanced green fluorescent protein; KDEL, endoplasmic reticulum retention signal KDEL; Nrxn, neurexin; SBP, streptavidin-binding protein; WT, wild type.
(AVI)

**S2 Movie. DIV9 WT rat cortical neuron co-expressing the ER hook (Streptavidin-KDEL) and SBP-EGFP-Nrxn1α (grayscale, inverted for clarity) and live-stained for the AIS marker Neurofascin to label the axon.** Cell was incubated with biotin for 22 min and then recorded every second for 120 s. The axon is indicated. Frame rate: 4 fps. AIS, axon initial segment; DIV, days in vitro; ER, endoplasmic reticulum; EGFP, enhanced green fluorescent protein; KDEL, endoplasmic reticulum retention signal KDEL; Nrxn, neurexin; SBP, streptavidin-binding protein; WT, wild type.
(AVI)

**S3 Movie. DIV10 WT rat cortical neuron co-expressing the ER hook (Streptavidin-KDEL) and SBP-EGFP-Nrxn1α (grayscale, inverted for clarity) and live-stained for the AIS marker Neurofascin to label the axon.** Cell was incubated with biotin for 1 h and 55 min and then recorded every second for 120 s. The axon is indicated. Frame rate: 4 fps. AIS, axon initial segment; DIV, days in vitro; ER, endoplasmic reticulum; EGFP, enhanced green fluorescent protein; KDEL, endoplasmic reticulum retention signal KDEL; Nrxn, neurexin; SBP, streptavidin-binding protein; WT, wild type.
(AVI)

**S4 Movie. DIV9 WT rat cortical neuron co-expressing the ER hook (Streptavidin-KDEL) and TfR-SBP-EGFP (grayscale, inverted for clarity) and live-stained for the AIS marker Neurofascin to label the axon.** Biotin was added 10 min after the beginning of the imaging session. Cell was recorded every 5 min for 2.5 h. The axon is indicated. Frame rate: 2 fps. AIS, axon initial segment; DIV, days in vitro; ER, endoplasmic reticulum; EGFP, enhanced green fluorescent protein; KDEL, endoplasmic reticulum retention signal KDEL; SBP, streptavidin-binding protein; TfR, transferrin receptor; WT, wild type.
(AVI)

**S5 Movie. DIV10 WT rat cortical neuron co-expressing the ER hook (Streptavidin-KDEL) and SBP-EGFP-Nrxn1α (grayscale, inverted for clarity) and live-stained for the AIS marker Neurofascin to label the axon.** Cell was recorded every second for 30 s. The axon is indicated. Frame rate: 4 fps. AIS, axon initial segment; DIV, days in vitro; ER, endoplasmic reticulum; EGFP, enhanced green fluorescent protein; KDEL, endoplasmic reticulum retention signal KDEL; Nrxn, neurexin; SBP, streptavidin-binding protein; WT, wild type.
(AVI)

**S6 Movie. DIV3 WT rat cortical neuron co-expressing the ER hook (Streptavidin-KDEL) and SBP-EGFP-Nrxn1α (grayscale, inverted for clarity) and live-stained for the AIS marker Neurofascin to label the axon.** Biotin was added 10 min after the beginning of the imaging session. Cell was recorded every 5 min for 2.5 h. The axon is indicated. Frame rate: 2 fps. AIS, axon initial segment; DIV, days in vitro; ER, endoplasmic reticulum; EGFP, enhanced green fluorescent protein; KDEL, endoplasmic reticulum retention signal KDEL; Nrxn,

neurexin; SBP, streptavidin-binding protein; WT, wild type.

(AVI)

**S1 Fig. *Sorcs1*$^{HA}$ KI mouse generation and SorCS1 regulates axonal surface polarization of Nrxn1α.** (A) CRISPR/Cas9-mediated generation of *Sorcs1*$^{HA}$ KI mice. Orange boxes represent the left and right homology arms. Blue box represents the ssDNA donor oligonucleotide containing the HA tag. Schematic representation of SorCS1 protein domain organization is shown to illustrate the HA-tagging of HA-SorCS1 downstream of the second furin cleavage site, right before the VPS10P domain (at the amino acid position 144). (B) Detection of HA-SorCS1 by western blot in total brain extracts prepared from *Sorcs1*$^{HA}$ KI mice (P60). Total protein staining (Ponceau) shows equal loading between lanes. See S9 Fig for raw uncropped blots. (C) High-zoom images of dendritic internalized (int.) and surface (s.) SorCS1 from DIV9 WT mouse hippocampal neurons expressing HA-tagged WT SorCS1cβ (WT) or Y1132A for 48 h. Live neurons were incubated with an anti-HA antibody and pulse-chased for 20 min. Neurons were immunostained for surface HA-SorCS1 (grayscale and green), internalized SorCS1 (grayscale and red) and MAP2 (blue). (D) Quantification of panel C: internalized SorCS1 fluorescence intensity relative to total levels and normalized to cells expressing WT-SorCS1, surface SorCS1 fluorescence intensity relative to total levels and normalized to cells expressing WT-SorCS1. WT ($n = 28$ neurons); Y1132A ($n = 30$). $^{***}P < 0.001$ (Mann-Whitney test, 3 independent experiments). (E) DIV10 *Sorcs1*$^{flox/flox}$ cortical neurons electroporated with EGFP (Ctr) or Cre-EGFP immunostained for pan-Nrxn (grayscale). (F) Quantification of panel E: pan-Nrxn fluorescence intensity in axon and dendrites normalized to cells expressing EGFP and ratio of axonal–dendritic pan-Nrxn intensity. Ctr ($n = 30$ neurons); Cre ($n = 29$). $^{*}P < 0.05$; $^{**}P < 0.01$ (Mann-Whitney test, 3 independent cultures). (G) Representative images of DIV8, DIV10 WT mouse cortical neurons electroporated with L315 control construct (Control) or L315 Nrxn triple knockdown construct (Nrxns TKD) immunostained for pan-Nrxn (grayscale) and EGFP (grayscale). Blue asterisk marks the cell body. (H) Quantification of panel G: Nrxn fluorescence intensity normalized to cells expressing EGFP ($n = 45$ neurons for each group). $^{***}P < 0.001$ (Mann-Whitney test, 3 independent experiments). (I) High-zoom images of Nrxn1α surface distribution from DIV8 to DIV9 WT mouse hippocampal neurons co-expressing extracellular HA-tagged Nrxn1α and an empty vector, HA-Nrxn1α and WT SorCS1-myc or HA-Nrxn1α and an endocytosis-defective mutant of SorCS1cβ-myc (Y1132A) for 48 h. Neurons were immunostained for surface HA-Nrxn1α (grayscale and green) and MAP2 (blue). (J) Quantification of panel I: surface HA-Nrxn1α fluorescence intensity in axon and dendrites relative to total surface levels and normalized to cells expressing the empty vector and ratio of axonal–dendritic surface HA intensity. $^{**}P < 0.01$; $^{***}P < 0.001$ (Kruskal-Wallis test followed by Dunn's multiple comparisons test, 4 independent experiments; $n = 40$ neurons for each group). Underlying numerical values can be found in S1 Data. Graphs show mean ± SEM. Scale bar, 5 μm (panels C and I), 20 μm (panels E and G). Cre, Cre recombinase; Ctr, control; DIV, days in vitro; EGFP, enhanced green fluorescent protein; HA, hemagglutinin; KI, knock in; LRD, leucine-rich domain; MAP2, microtubule-associated protein 2; Nrxn, neurexin; PKD, polycystic kidney disease domain; Pro, propeptide; SorCS1, Sortilin-related CNS expressed 1; ssDNA, single-strand DNA; SP, signal peptide; TKD, triple knock down; TM, transmembrane; VPS10P, vacuolar protein sorting 10 protein; WT, wild type.

(TIF)

**S2 Fig. *Nrxn1α*$^{HA}$ KI mouse generation and Nrxn1α localizes to the somatodendritic and axonal compartment.** (A) Orange boxes represent the left and right homology arms. Blue box represents the ssDNA donor oligonucleotide containing the HA tag. CRISPR/

Cas9-mediated HDR allowed for precise HA-tagging of the *Nrxn1* locus right after the SP. An AatII restriction site was introduced in the DNA sequence coding for the HA tag, by taking advantage of the redundancy of the genetic code, in order to facilitate the distinction between homozygous and heterozygous *Nrxn1α*$^{HA}$ mice. Schematic representation of Nrxn1α protein domain organization is shown to illustrate the HA-tagging of HA-Nrxn1α right after the SP in the extracellular domain. (B) Left panels: detection of HA-Nrxn1α by western blot in cortical extracts prepared from adult *Nrxn1α*$^{HA}$ KI mice (3 months old). Total protein staining by using Ponceau method was used as loading control. Asterisk indicates an aspecific band. Right panel: identification of WT (+/+), heterozygous (+/Tg), and homozygous (Tg/Tg) *Nrxn1α*$^{HA}$ mice by generation of a PCR fragment of 499 bp followed by AatII-mediated restriction endonuclease reaction generating 2 DNA fragments of 312 bp and 187 bp. See S9 Fig for raw uncropped blots. (C and D) Representative images of permeabilized DIV7 *Nrxn1α*$^{HA}$ mouse cortical neurons cultured together with WT mouse cortical neurons and immunostained for HA-Nrxn1α (green and grayscale) and MAP2 (blue). Red arrowheads indicate the axon, and the blue asterisk marks the cell body. High-zoom images of the subcellular distribution of endogenous HA-Nrxn1α: cell body (1), dendrites (2) and axon (3 and 3′). (E) Permeabilized DIV3, DIV7, and DIV11 WT mouse cortical neurons labeled with a pan-Nrxn antibody (grayscale). MAP2 (blue) and Ankyrin-G (AnkG; red) identify somatodendritic and axonal compartments, respectively. (F) Quantification of panel E: pan-Nrxn fluorescence intensity in axon and dendrites normalized to DIV3 neurons and ratio of axonal–dendritic pan-Nrxn intensity. DIV3 (*n* = 30 neurons); DIV7 (*n* = 30); DIV11 (*n* = 28). **$P < 0.01$; ***$P < 0.001$ (Kruskal-Wallis test followed by Dunn's multiple comparisons test, 3 independent cultures). Underlying numerical values can be found in S1 Data. Graphs show mean ± SEM. Scale bars, 20 μm (panels C, D, and E); 10 μm (panel D [high-zoom]). AatII, AatII resctriction enzyme; AnkG, Ankyrin-G; DIV, days in vitro; EGF, epidermal growth factor-like; HA, hemagglutinin; HDR, homology directed repair; KI, knock in; LNS, laminin/neurexin/sex hormone-binding globulin; MAP2, microtubule-associated protein 2; Nrxn, neurexin; PCR, polymerase chain reaction; SP, signal peptide; ssDNA, single-strand DNA; TM, transmembrane; WT, wild type. (TIF)

**S3 Fig. Delayed axonal trafficking of Nrxn1α in mature neurons.** (A) Illustration of the strategy used during live-cell imaging experiments to label the axon. Representative image of a live DIV8 WT rat cortical neuron co-expressing SBP-EGFP-Nrxn1α (grayscale and green) and streptavidin-KDEL immunostained for pan-Neurofascin (axon initial segment marker, grayscale and red). (B) Quantification of SBP-EGFP-Nrxn1α (*n* = 23 neurons) fluorescence intensity in soma, dendrites and axon in 3 independent experiments. (C and D) Mean number of SBP-EGFP-Nrxn1α vesicles moving in anterograde and retrograde direction in (C) dendrites and (D) axons. In dendrites, analysis was performed at 2 different time intervals: 20 to 25 min (*n* = 8 neurons), and 1 to 2 h (*n* = 16) after adding biotin; in axons, analysis was performed 1 to 2 h after biotin (*n* = 16). *$P < 0.05$; **$P < 0.01$; ***$P < 0.001$ (Mann-Whitney test). (E) Live-cell imaging in DIV8 to DIV10 WT rat cortical neurons co-expressing TfR-SBP-EGFP and ER hook. After 24 to 31 h of expression, neurons were imaged every 5 min for 2.5 h. Biotin was added 10 min after the beginning of the imaging session. Shown are representative images of TfR-SBP-EGFP fluorescence in dendrites and axon before (t0) and 30, 60, 90, and 120 min after adding biotin. Red arrowheads indicate axon, and black arrows indicate TfR-SBP-EGFP-positive puncta. See also S4 Movie. (F) Quantification of TfR-SBP-EGFP (*n* = 13 neurons) fluorescence intensity in soma, dendrites and axon in 3 independent experiments. (G) Live-cell imaging in DIV3 WT rat cortical neurons co-expressing SBP-EGFP-Nrxn1α and streptavidin-KDEL. After 25 to 29 h of expression, neurons were imaged every 5 min for 2.5 h. Biotin

was added 10 min after the beginning of the imaging session. Shown are representative images of EGFP-Nrxn1α endogenous fluorescence in dendrites and axons before (t0), 30, 60, 90, and 120 min after adding biotin. Red arrowheads indicate the axon and black arrows indicate EGFP-Nrxn1α-positive puncta. See also S6 Movie. (H) Quantification of EGFP-Nrxn1α fluorescence intensity in the soma, dendrites and axons ($n = 9$ neurons) in 2 independent experiments. Underlying numerical values can be found in S1 Data. Graphs show mean ± SEM. Scale bars, 20 μm (panel A); 10 μm (panels E and G). DIV, days in vitro; ER, endoplasmic reticulum; EGFP, enhanced green fluorescent protein; KDEL, endoplasmic reticulum retention signal KDEL; Nrxn, neurexin; SBP, streptavidin-binding protein; TfR, transferrin receptor; WT, wild type.
(TIF)

**S4 Fig. Nrxn1α is internalized from the dendritic surface and incorporated into endosomes.** (A–I) Antibody pulse-chase experiments in DIV8 to DIV11 WT mouse cortical neurons expressing extracellular HA-tagged Nrxn1α or co-expressing HA-Nrxn1α and endosomal markers for 48 h. (A) Representative images of mouse cortical neurons pulse-chased for 15 min and immunostained for internalized HA-Nrxn1α (grayscale and green), MAP2 (blue), and Ankyrin-G (red). Red arrowheads indicate the axon, and the blue asterisk marks indicate the cell body. (B) Quantification of internalized Nrxn1α fluorescence intensity in dendrites and axons relative to total levels ($n = 15$ neurons). $^{***}P < 0.001$ (Mann-Whitney test, 3 independent experiments). Graph shows mean ± SEM. (C–H) High-zoom images of dendritic segments from mouse cortical neurons pulse-chased for 15 min or 40 min and immunostained for internalized HA-Nrxn1α (grayscale and green), MAP2 (blue); EEA1 (EE marker, in grayscale and red) (C), EGFP-Rab5 (EE marker, in grayscale and red) (D), EGFP-Rab11 (RE marker, in grayscale and red) (E), EGFP-Rab7 (late endosomal marker, in grayscale and red) (F), LAMP1 (lysosomal marker, in grayscale and red) (G) or SorCS1-myc (in grayscale and red) (H). (I) Quantification of the colocalization of internalized Nrxn1α with different endosomal or lysosomal markers expressed as Mander's coefficient ($n = 15$ neurons for each group). Graph shows mean. $^{*}P < 0.05$; $^{**}P < 0.01$ (Mann-Whitney test, 3 independent experiments). Underlying numerical values can be found in S1 Data. Scale bars, 20 μm (panel A); 5 μm (panel H). DIV, days in vitro; EE, early endosome; EEA1, early endosome antigen 1; HA, hemagglutinin; LAMP1, lysosomal-associated membrane protein 1; MAP2, microtubule-associated protein 2; Nrxn, neurexin; Rab, Rab GTPase; RE, recycling endosome; SorCS1, Sortilin-related CNS expressed 1; WT, wild type.
(TIF)

**S5 Fig. Endocytosis and transport via endosomes are required to sustain axonal levels of Nrxn1α.** (A) Representative images of DIV8 to DIV10 WT mouse cortical neurons co-expressing HA-Nrxn1α and EGFP-tagged WT Dynamin1 (Ctr) or a dominant negative of EGFP-Dynamin1 (K44A, to block Dynamin-dependent endocytosis) for 48 h. Neurons were immunostained for surface HA-Nrxn1α in grayscale. Red arrowheads indicate the axon, and the blue asterisk marks the cell body. High-zoom images of dendritic (D, dotted blue box) and axonal (A, dotted red box) Nrxn1α are shown next to the whole-cell images. (B) Quantification of panel A: surface HA-Nrxn1α fluorescence intensity in axon and dendrites relative to total surface levels and normalized to cells expressing Dyn.1 WT and ratio of axonal–dendritic surface HA intensity. Ctr ($n = 30$ neurons); K44A ($n = 30$). (C) Representative images of DIV8 to DIV9 WT mouse cortical neurons expressing extracellular HA-tagged Nrxn1α and treated either with DMSO (vehicle, Ctr) or with Dynasore (Dyn) 30 h after transfection. Neurons were immunostained 18 h after treatment for surface HA-Nrxn1α (grayscale). (D) Quantification of panel C; Ctr ($n = 29$ neurons); Dyn ($n = 25$). (E) Representative images of DIV8 to

DIV10 WT cortical neurons co-expressing HA-Nrxn1α and EGFP-tagged WT Rab5 (Ctr) or a dominant negative of EGFP-Rab5 (S34N, to prevent the formation of EEs) for 48 h. Neurons were immunostained for surface HA-Nrxn1α (grayscale). (F) Quantification of panel E; Ctr ($n$ = 29 neurons); S34N ($n$ = 27). (G) Representative images of DIV8 to DIV10 WT cortical neurons co-expressing HA-Nrxn1α and EGFP-tagged WT Rab11 (Ctr) or a dominant negative of EGFP-Rab11 (S25N, to prevent the formation of REs) for 48 h. Neurons were immunostained for surface HA-Nrxn1α (grayscale). (H) Quantification of panel G; Ctr ($n$ = 30 neurons); S25N ($n$ = 26). (I) Representative images of DIV8 to DIV10 WT mouse cortical neurons co-expressing extracellular HA-tagged Nrxn1α and EGFP-tagged WT Rab7 (Ctr) or a dominant negative of EGFP-Rab7 (T22N, to prevent the formation of late endosomes) for 48 h. Neurons were immunostained for surface HA-Nrxn1α (grayscale). (J) Quantification of panel I; Ctr ($n$ = 30 neurons); T22N ($n$ = 29). *$P$ < 0.05; **$P$ < 0.01; ***$P$ < 0.001 (Mann-Whitney test, 3 independent cultures). Underlying numerical values can be found in S1 Data. Graphs show mean ± SEM. Scale bars, 20 μm (whole-cell panels); 5 μm (high-zoom images). Ctr, control; DIV, days in vitro; EE, early endosome; EGFP, enhanced green fluorescent protein; HA, hemagglutinin; Nrxn, neurexin; Rab, Rab GTPase; RE, recycling endosome; WT, wild type.
(TIF)

**S6 Fig. Selective missorting of axonal cargo in the absence of SorCS1.** (A) High-zoom representative images of dendritic internalized Nrxn1α from DIV8, DIV10 *Sorcs1*$^{flox/flox}$ cortical neurons electroporated with EGFP (Control) or Cre-EGFP (Cre) and transfected with extracellular HA-tagged Nrxn1α. After 48 h, neurons were live-labeled with anti-HA antibody and pulse-chased for 20 min, followed by immunostaining for internalized HA-Nrxn1α (grayscale and green) and MAP2 (grayscale and blue). (B) Quantification of panel A: internalized Nrxn1α fluorescence intensity relative to total levels and normalized to cells expressing EGFP (3 independent experiments). Control ($n$ = 30 neurons); Cre ($n$ = 28). (C and D) Representative images of DIV8 to DIV10 *Sorcs1*$^{flox/flox}$ mouse cortical neurons electroporated with EGFP (Ctr) or Cre-EGFP, and transfected with myc-L1 (C) or HA-Caspr2 (D) for 48 h, followed by immunostaining for surface myc-L1 or HA-Caspr2 in grayscale. (E) Quantification of panels C and D: surface myc-L1 and HA-Caspr2 fluorescence intensity in axon and dendrites relative to total surface levels and normalized to cells expressing EGFP and ratio of axonal–dendritic surface L1 and Caspr2 intensity. Ctr_L1 ($n$ = 29 neurons); Cre_L1 ($n$ = 26); Ctr_Caspr2 ($n$ = 28); Cre_Caspr2 ($n$ = 25). *$P$ < 0.05; **$P$ < 0.01; ***$P$ < 0.001 (Mann-Whitney test, 4 independent cultures). Underlying numerical values can be found in S1 Data. Graphs show mean ± SEM. Scale bars, 5 μm (panel A), 20 μm (panel D). Caspr2, contactin-associated protein-like 2; Cre, Cre recombinase; Ctr, control; DIV, days in vitro; EGFP, enhanced green fluorescent protein; HA, hemagglutinin; MAP2, microtubule-associated protein 2; Nrxn, neurexin; SorCS1, Sortilin-related CNS expressed 1.
(TIF)

**S7 Fig. Sorting of somatodendritic cargo is not affected by loss of SorCS1.** (A) Representative images of DIV14 *Sorcs1*$^{flox/flox}$ cortical neurons co-expressing EGFP-GluA2 and an empty vector (Ctr) or EGFP-GluA2 and Cre-myc (Cre) for 7 d. Neurons were immunostained for EGFP-GluA2 (grayscale). Red arrowheads indicate the axon, and the blue asterisk marks the cell body. (B) Representative images of DIV10 mouse *Sorcs1*$^{flox/flox}$ cortical neurons electroporated with EGFP (Ctr) or Cre-EGFP (Cre) immunostained for MAP2 (grayscale). (C) Quantification of panels A and B: dendritic versus axonal distribution (D:A–polarity index) of GluA2 or MAP2 in WT and *Sorcs1* KO cells (at least 1 independent experiment). GluA2_Ctr ($n$ = 9 neurons); GluA2_Cre ($n$ = 9); MAP2_Ctr ($n$ = 18); MAP2_Cre ($n$ = 20). Underlying numerical values can be found in S1 Data. Graph shows mean ± SEM. Scale bars, 20 μm. Cre, Cre

recombinase; Ctr, control; DIV, days in vitro; EGFP, enhanced green fluorescent protein; GluA2, glutamate ionotropic receptor AMPA type subunit 2; KO, knock out; MAP2, microtubule-associated protein 2; SorCS1, Sortilin-related CNS expressed 1; WT, wild type.
(TIF)

**S8 Fig. Effect of deletion mutants on Nrxn1α surface polarization.** (A) Schematic representation of WT and C-terminal mutants of rat Nrxn1α. Different deletion mutants of Nrxn1α C terminal were designed: deletion mutant of the entire C terminal (ΔC), deletion mutant spanning the regions 1495–1511 and 1516–1527 (ΔA), deletion mutant lacking a putative phosphorylation site in serine 1513 (ΔS), deletion mutant lacking the PDZ-binding motif (ΔP), and deletion mutant lacking the 4.1- binding motif (Δ4.1). (B) Representative images of DIV8 to DIV10 WT mouse cortical neurons expressing extracellular HA-tagged WT-Nrxn1α and C-terminal deletions mutants of Nrxn1α for 48 h and immunostained for surface HA-Nrxn1α (grayscale). (C) Quantification of panel B: surface HA-Nrxn1α fluorescence intensity in axon and dendrites relative to total surface levels and normalized to cells expressing WT-Nrxn1α and ratio of axonal–dendritic surface HA intensity. WT ($n$ = 40 neurons); ΔC ($n$ = 30); ΔA ($n$ = 30); ΔS ($n$ = 28), ΔP ($n$ = 28). ***$P < 0.001$ (Kruskal-Wallis test followed by Dunn's multiple comparisons test, at least 3 independent experiments). (D) Representative images of DIV8 to DIV10 WT mouse cortical neurons expressing extracellular HA-tagged WT-Nrxn1α and N-terminal deletion mutant of Nrxn1α for 48 h and immunostained for surface HA-Nrxn1α (grayscale). (E) Quantification of panel D: surface HA-Nrxn1α fluorescence intensity in axon and dendrites relative to total surface levels and normalized to cells expressing WT-Nrxn1α and ratio of axonal–dendritic surface HA intensity. WT ($n$ = 30 neurons); ΔNT ($n$ = 29). ***$P < 0.001$ (Mann-Whitney test, 3 independent experiments). Underlying numerical values can be found in S1 Data. Graph shows mean ± SEM. Scale bars, 20 μm. DIV, days in vitro; HA, hemagglutinin; Nrxn, neurexin; PDZ, PSD-95/discs large/zona occludens-1 motif; WT, wild type.
(TIF)

**S9 Fig. Raw uncropped blots.** Raw uncropped blots for Fig 3I, S1B Fig, and S2B Fig.
(TIF)

**S1 Data. Numerical data underlying Figs 1 through 5 and S1 through S8 Figs.**
(XLSX)

**S1 Table. List of plasmids used in this study.**
(XLSX)

**S2 Table. List of antibodies used in this study.**
(XLSX)

## Acknowledgments

We thank Anders Nykjaer, Pierre Vanderhaeghen, Sara Calafate, Heather Rice, Elsa Lauwers, Ragna Sannerud, and Lucía Chávez-Gutiérrez for critical reading of the manuscript and de Wit lab members for discussion and comments. We thank Casper Hoogenraad for experimental advice and reagents, and Ragna Sannerud, Catherine Faivre-Sarrailh, Rytis Prekeris, Juan S. Bonifacino, Jeremy Tavare, Lorna Hodgson, Dan P. Felsenfeld, Takeshi Sakurai, Marco Arese, Thomas C. Südhof, Peter Scheiffele, Thomas Biederer, Eunjoon Kim, Gopal Thinakaran, Ryohei Iwata, Aaron Bowen, and Matthew Kennedy for reagents. We thank Lutgarde Serneels for

advice on CRISPR/Cas9-based KI mouse generation, and Pieter Vanden Berghe and Valérie Van Steenbergen for live-cell imaging analysis advice and sharing Igor-based software.

## Author Contributions

**Conceptualization:** Luís F. Ribeiro, Joris de Wit.

**Investigation:** Luís F. Ribeiro, Ben Verpoort, Julie Nys, Kristel M. Vennekens, Keimpe D. Wierda.

**Supervision:** Joris de Wit.

**Writing – original draft:** Luís F. Ribeiro, Joris de Wit.

**Writing – review & editing:** Luís F. Ribeiro, Joris de Wit.

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
