## [Editor Report · Decision Letter 0]

8 Jul 2019

Dear Dr De Wit, 

Thank you for submitting your manuscript entitled "SorCS1-mediated sorting in dendrites maintains neurexin axonal surface polarization required for synaptic function" for consideration as a Research Article by PLOS Biology.

Your manuscript has now been evaluated by the PLOS Biology editorial staff, as well as by an Academic Editor with relevant expertise, and I am writing to let you know that we would like to send your submission out for external peer review.

**Important**: Please also see below for further information regarding completing the MDAR reporting checklist. The checklist can be accessed here: https://plos.io/MDARChecklist

Please re-submit your manuscript and the checklist, within two working days, i.e. by Jul 10 2019 11:59PM.

Kind regards,

Gabriel Gasque, Ph.D.,

Senior Editor

PLOS Biology

INFORMATION REGARDING THE REPORTING CHECKLIST:

PLOS Biology is pleased to support the "minimum reporting standards in the life sciences" initiative (https://osf.io/preprints/metaarxiv/9sm4x/). This effort brings together a number of leading journals and reproducibility experts to develop minimum expectations for reporting information about Materials (including data and code), Design, Analysis and Reporting (MDAR) in published papers. We believe broad alignment on these standards will be to the benefit of authors, reviewers, journals and the wider research community and will help drive better practise in publishing reproducible research. 

We are therefore participating in a community pilot involving a small number of life science journals to test the MDAR checklist. The checklist is intended to help authors, reviewers and editors adopt and implement the minimum reporting framework. 

IMPORTANT: We have chosen your manuscript to participate in this trial. The relevant documents can be located here:

MDAR reporting checklist (to be filled in by you): https://plos.io/MDARChecklist

**We strongly encourage you to complete the MDAR reporting checklist and return it to us with your full submission, as described above. We would also be very grateful if you could complete this author survey:

https://forms.gle/seEgCrDtM6GLKFGQA

Additional background information:

Interpreting the MDAR Framework: https://plos.io/MDARFramework

Please note that your completed checklist and survey will be shared with the minimum reporting standards working group. However, the working group will not be provided with access to the manuscript or any other confidential information including author identities, manuscript titles or abstracts. Feedback from this process will be used to consider next steps, which might include revisions to the content of the checklist. Data and materials from this initial trial will be publicly shared in September 2019. Data will only be provided in aggregate form and will not be parsed by individual article or by journal, so as to respect the confidentiality of responses. 

Please treat the checklist and elaboration as confidential as public release is planned for September 2019.

We would be grateful for any feedback you may have.

---

## [Decision Letter · Decision Letter 1]

20 Aug 2019

Dear Joris,

Thank you very much for submitting your manuscript "SorCS1-mediated sorting in dendrites maintains neurexin axonal surface polarization required for synaptic function" for consideration as a Research Article by PLOS Biology. Your paper was evaluated by the PLOS Biology editors as well as by an Academic Editor with relevant expertise and by three independent reviewers. Based on the reviews, we will probably accept this manuscript for publication, providing that you will modify the manuscript to address the comments and questions raised by the reviewers. Please accept my apologies for the delay in sending this decision to you.

We expect to receive your revised manuscript within two weeks. Your revisions should address the specific points made by each reviewer. Please submit a file detailing your responses to the editorial requests and a point-by-point response to all of the reviewers' comments that indicates the changes you have made to the manuscript. In addition to a clean copy of the manuscript, please upload a 'track-changes' version of your manuscript that specifies the edits made.

In addition to the remaining revisions and before we will be able to formally accept your manuscript and consider it "in press", we also need to ensure that your article conforms to our guidelines and policies. Please read below my signature for **IMPORTANT** information regarding your Data Availability Statement.

A member of our team will be in touch shortly with a set of requests. As we can't proceed until these requirements are met, your swift response will help prevent delays to publication.

******* 

Please note that you may have the opportunity to make the peer review history publicly available. The record will include editor decision letters (with reviews) and your responses to reviewer comments. If eligible, we will contact you to opt in or out.

Early Version

Sincerely,

Gabriel Gasque, Ph.D., 

Senior Editor

PLOS Biology

ETHICS STATEMENT:

The Ethics Statements in the submission form and Methods section of your manuscript should match verbatim. Please ensure that any changes are made to both versions.

DATA POLICY:

Note that we do not require all raw data. Rather, we ask for all individual quantitative observations that underlie the data summarized in the figures and results of your paper. For an example see here: http://www.plosbiology.org/article/info%3Adoi%2F10.1371%2Fjournal.pbio.1001908#s5

These data can be made available in one of the following forms:

Regardless of the method selected, please ensure that you provide the individual numerical values that underlie the summary data displayed in the following figure panels: Figures 1CE, 2BGI, 3BDFHM, 4BDFHJ, 5BDFGHJK, S1DFHJ, S2F, S3ABCDG, S4BDFHJL, S5BE

Please also note that PLOS does not permit references to “data not shown.” You should provide the relevant data within the manuscript, the Supporting Information files, or in a public repository. If the data are not a core part of the research study being presented, we ask that you remove any references to these data. In this case, some of the data not shown seem relevant to your conclusions, including:

“The polarity index of two somatodendritic proteins (GluA2-GFP and MAP2), was unchanged in Sorcs1 KO neurons (data not shown), indicating that loss of SorCS1 does not generally perturb the polarized distribution of neuronal proteins.”

“We systematically tested the effect of a series of Nrxn1a cytoplasmic deletions on surface polarization (data not shown) and found that removing the 4.1-binding motif in the cytoplasmic domain (Nrxn1a �4.1) was the only deletion that dramatically increased Nrxn1a axonal surface polarization (Figure 4A, B).”

“We find that the extracellular domain of Nrxn1a is required for somatodendritic sorting (data not shown), suggesting that deletion of the 4.1-binding motif in Nrxn1a impairs dendritic insertion but not initial sorting to this compartment.”

Thus, please include these data.

The references to antibodies not shown in the figure legends possibly can be omitted. 

Please ensure that the figure legends in your manuscript include information on where the underlying data can be found.

For manuscripts submitted on or after 1st July 2019, we require the original, uncropped and minimally adjusted images supporting all blot and gel results reported in an article's figures or Supporting Information files. We will require these files before a manuscript can be accepted so please prepare them now, if you have not already uploaded them. Please carefully read our guidelines for how to prepare and upload this data: https://journals.plos.org/plosbiology/s/figures#loc-blot-and-gel-reporting-requirements.

Reviewer remarks:

Reviewer's Responses to Questions

Reviewer #1: Ribeiro et al. describe a role for SorCS1 in the transcytosis of neurexin from the dendrite to the axon. The idea of transcytosis for axonal polarization is not new, but the molecular details required for this process are presently unknown. This study provides great mechanistic insight and follows the life journey of an important neuronal receptor. The authors show how an axonal cargo first transits to the dendrite where it is endocytosed and then resorted via SorCS1 and Rip11 from early to recycling endosomes to be polarized toward the axon. This is a very comprehensive study that using cutting edge techniques such as CRISPR knockins and dynamic imaging. The detailed cell biological studies and data analysis provide a rigorous and thorough understanding of the described transcytosis mechanism.

This manuscript represents a conceptual advance and significantly advances our knowledge of polarity neuronal membrane trafficking processes. There are no major technical concerns and only one question and a few minor points. 

Questions:

Can the authors describe the localization of Neurexin relative to other endosomal markers (as shown in Fig. 3) for additional time points than those shown? I am not proposing to show all the time points for all the RABs. But it seems that they should mention whether the time point shown are the only ones a particular RAB showed SorCS1 dependent behavior.

Minor points:

1. There is a type on Figure 2 (RE lumen instead of ER lumen)

2. The text and Figure S3 refer to neurofascin as the axonal marker, but the corresponding main figure (Fig 2D) shows AnkG.

3. The first two sentences of the ‘Nrxn1α is transcytosed’ section are confusing based on how it is presently written. The previous section showing the RUSH data seems to answer the question asked in these two sentences.

Reviewer #2: Ribeiro et al. report a detailed analysis of alpha-Neurexin-1 (aNrxn1) trafficking and the regulation by SOR-CS1 in cortical neurons. Based on previous work identifying the function of SOR-CS1 protein as sorting receptor for AMPARs and Nrxn1 and Nrxn2 (Savas et al. 2015) the authors here report the specific regulation of the axonal sorting of aNrxn1 by SOR-CS1 and new identified interaction partners as Rip11. 

Probing the function of SOR-CS1 at different developmental time points allowed to identify the impact of SOR-CS1 in the recycling of aNrxn-1 on the dendritic membrane and sorting in recycling endosomes for axonal trafficking. A long standing issue in analysing the surface distribution and sorting of adhesion molecules in neurons, particular aNrxns, is the lack of specific antibodies against their extracellular domains. The authors overcome this by the use of a newly generated knock-in mouse of HA-tagged aNrxn-1. In combination with a second knock-in mouse for HA-tagged SOR-CS1 the core results of the work are very convincing, supporting the function of SOR-CS1 for the regulation of the dendritic expression of aNrxn1 and retargeting to the axon, as well as the functional impact of this sorting process on neurotransmission and short-term synaptic plasticity. The paper and the here introduced tools are well suited to explore the discussed function of dendritic expressed aNrxn1 as regulator of adhesion contacts and synapse formation as well as the presynaptic actions of aNrxn1. 

Minor points:

In respect to the previous work, the transcytosis of aNrxn1 in cortical neurons seem to be the major process in synapse formation and function and does only little affect postsynaptic AMPAR-density and function. The formation and function of synapses depend on the here investigated trafficking of aNrxn1 but seem not being affected by postsynaptic changes in AMPAR trafficking, as proposed by Savas et al. 2015. Indeed, the probed parameters indicate a strong effect of SOR-CS1 KO on presynaptic functions, but are their remaining increased numbers of postsynaptic silent synapses, as discussed? May the autaptic cultures bias the effect on AMPARs by homeostatic upregulation of receptor density? Are their data for the surface population of AMPARs in these autaptic cultures, which could support the idea of postsynaptic AMPAR-silent synapses. 

The mentioned dendritic sorting motives in page 13, second paragraph are probably wrong spelled (“YxxO and [D/E]xxx[L/I] dendritic sorting motifs [48] but….”)

Reviewer #3: In this manuscript, Ribeiro et al examined the role of a VPS 10 family member sorting receptor SorCS1 in trafficking Nrxn, a presynaptic adhesion molecule and in synaptic transmission. Previously they have shown an interaction between these two proteins and a role of SorCS1 in Nrxn trafficking. Using the CRISPR/Cas9 technique, they generated two knock-in mouse line with endogenous tagged Nrxn or SorCS1 in order to study the localization and trafficking of these proteins at endogenous levels. Furthermore, they used RUSH system to visualize the trafficking of engineered Nrxn in a synchronized manner using live imaging. They also identified Rip11 as a binding partner for SorCS1 on endosome and showed that Rip11 is required for SorCS1 mediated transition of internalized Nrxn1α from early to recycling endosomes and facilitates Nrxn1α surface polarization toward the axon. Overall, this manuscript is well written. The authors have done tremendous amount of work and the experiments are well designed. The data is analyzed carefully and presented beautifully. I only have a few minor comments:

1. Fig. 3E, why the number of Nrxn punta is so few compared to Fig. 3A and 3C in the Cre expressing conditions? In Fig. 3G, is the number of Nrxn punta decreased in Cre expressing conditions in axons? 

2. Is the expression of SorCS1 or Rip11 developmentally regulated?

---

## [Editor Report · Decision Letter 2]

8 Oct 2019

Dear Dr De Wit,

On behalf of my colleagues and the Academic Editor, Franck Polleux, I am pleased to inform you that we will be delighted to publish your Research Article in PLOS Biology. 

Early Version

PRESS 

Kind regards,

Sofia Vickers

Senior Publications Assistant

PLOS Biology

On behalf of, 

Gabriel Gasque,

Senior Editor

PLOS Biology